# Different factors control long-term versus short-term outcomes for bacterial colonisation of a urinary catheter

Freya Bull [1,2], Sharareh Tavaddod [1], Nick Bommer[3], Meghan Perry [4], Chris A. Brackley [1] & Rosalind J. Allen [1,5,6] ✉

Urinary catheters are used extensively in hospitals and long-term care and they are highly prone to infection. Understanding the pathways by which bacteria colonise a urinary catheter could guide strategies to mitigate infection, but quantitative models for this colonisation process are lacking. Here we present a mathematical model for bacterial colonisation of a urinary catheter that integrates population dynamics and fluid dynamics. The model describes bacteria migrating up the outside surface of the catheter, spreading into the bladder and being swept through the catheter lumen. Computer simulations of the model reveal that clinical outcomes for long-term versus short-term catheterisation are controlled by different factors: the rate of urine production by the kidneys as opposed to urethral length, catheter surface properties and bacterial motility. Our work may help explain variable susceptibility to catheter-associated urinary tract infection (CAUTI) among individuals and the mixed success of antimicrobial surface coatings. Our model suggests that for long-term catheterised patients, increasing fluid intake or reducing residual urine volume in the bladder may help prevent infection, while antimicrobial surface coatings are predicted to be effective only for short-term catheterised patients. Therefore, different catheter management strategies could be rationally targeted to long-term vs short-term catheterised patients.

Urinary catheters—thin tubes which are inserted into the urethra to drain the bladder—are commonly used in hospitals, especially intensive care units, and in community care for patients with urological issues[1,2]. Approximately 30 million urinary catheters are used annually in the USA[2]. The duration of catheterisation varies from a few days ('short-term', e.g. in hospital, post-surgery) to up to 3 months, followed by replacement('long-term', e.g. in community care settings). Urinary catheters are highly prone to colonisation by bacteria. This leads initially to catheter-associated bacteriuria(bacteria in the urine), which may be asymptomatic, not requiring treatment[3]. However, bacteriuria

can progress to symptomatic catheter-associated urinary tract infection (CAUTI), with potential complications including localised issues such as catheter blockage, bladder inflammation and the systemic and potentially fatal complication of sepsis. Bacteriuria has incidence rates of 3–7% per catheter day[4,5] and is almost universal in long-term catheterised patients[6]; the estimated incidence rate of CAUTI is about ten times lower, at 0.3–0.7% per catheter day[3,7]. In hospital settings, the likelihood of progression of bacteriuria to CAUTI has been estimated at 24% (uncertainty in this figure is high due to limited data)[8]. CAUTI are associated with a very significant societal and economic burden[9,10]:

[1]School of Physics and Astronomy, University of Edinburgh, Edinburgh, UK. [2]Department of Mathematics, University College London, London, UK. [3]Veterinary Specialists Scotland, Livingston, UK. [4]Clinical Infection Research Group, Regional Infectious Diseases Unit, Western General Hospital, Edinburgh, UK. [5]Theoretical Microbial Ecology, Institute of Microbiology, Faculty of Biological Sciences, Friedrich Schiller University Jena, Jena, Germany. [6]Cluster of Excellence Balance of the Microverse, Friedrich Schiller University Jena, Jena, Germany. ✉e-mail: rosalind.allen@uni-jena.de

for example, they account for up to 40% of hospital-acquired infections[2,5,7,11–14]. Despite this, our understanding of the role of different factors in CAUTI development is limited[4,5,7,15–17]. In particular, it is not known why some patients are plagued by recurrent CAUTI while others are hardly troubled[18], why some infections develop far more rapidly than others[19], which of several possible infection sources is relevant for any given patient[7,20–22], and why efforts to prevent CAUTI using antimicrobial catheter coatings have shown mixed results in clinical trials[23,24]. To address these questions and to guide the rational design of more effective strategies to prevent or mitigate CAUTI[25], better understanding of the pathways by which bacteria colonise a urinary catheter is required.

Here we present a quantitative mathematical model for the dynamics of bacterial colonisation of a urinary catheter. Our study focuses on the widely-used indwelling Foley catheter, in which the catheter tip sits inside the bladder, held in place by a balloon, while the catheter tubing passes through the urethra, terminating at an external drainage bag (Fig. 1). We model the population dynamics of infecting bacteria as they migrate up the catheter, proliferate in the residual sump of urine that remains in the bladder during catheterisation[9,26–28], and travel out of the bladder in the urine flow through the catheter lumen, where they can attach to the inner lumen surface[5] and trigger biofilm formation that can block the catheter[7]. Our model allows us to predict the impact on clinical outcomes of different factors associated with the patient, the catheter and the infecting bacterial strain. These factors appear as numerical parameters in our model, for example, the length of the urethra or the rate of urine production. By running computer simulations of our model for different values of the various parameters, we predict the time course of bacterial colonisation of the catheter. Simulating the model for long times (several months) allows us to predict clinical outcomes for long-term catheterised patients, while simulations over short times (several days) can be used to mimic short-term catheterisation.

Over long times (long-term catheterisation), our model shows two distinct alternative outcomes with high versus low bacterial abundance in the urine, which we associate with bacteriuria versus no bacteriuria. Which of these clinical outcomes is obtained depends on the rate of urine production by the kidneys, as well as the volume of residual urine and the bacterial growth rate in urine. Varying the urine production rate in a range close to the typical human value can cause a transition between bacteriuria and no bacteriuria. This suggests a possible explanation for why some individuals are much more susceptible to catheter colonisation than others and points to increasing fluid intake as a potentially effective strategy to avoid bacteriuria in long-term catheterised patients. In contrast, for short-time simulations, mimicking short-duration catheterisation, the clinical outcome depends on whether the catheter is removed before bacteria have time to colonise the bladder. Therefore, the model prediction for the time before bacteriuria occurs is clinically relevant. Our model shows that this time is controlled by a different set of factors–the catheter length and the growth and mobility of bacteria on the catheter surface. This may help to explain why sex is a strong risk factor for CAUTI. This result also suggests that antimicrobial coatings, which alter bacterial growth dynamics on the catheter surface, may be effective for some, but not all, short-term catheterised patients, potentially helping to explain apparently contradictory results of clinical trials for antimicrobial coatings[23,24]. Our model also hints that the spatial pattern of bacterial biofilm growth on the catheter might provide clues about the source of CAUTI infections.

Innovative approaches to reduce the incidence of CAUTI are much needed, since there have been few effective improvements in urinary catheter design since the invention of the Foley catheter in the 1930s[9,27]. Our study indicates how mathematical models could help guide innovation. Specifically, our results suggest a need for targeted intervention strategies for different patient categories. For short-term catheterisation, bacteria on the catheter surface should be targeted, e.g. via antimicrobial surface coatings. For long-term catheterisation, however, interventions should instead focus on increasing urine production rate or reducing residual urine volume.

## Results

### Mathematical model for catheter colonisation

Our model consists of four parts: the outside surface of the catheter, the bladder, the urine flow within the catheter, and the luminal (inside) surface of the catheter (Fig. 1). We model bacterial growth and motility (where relevant) in each part of the model and we also allow bacteria to move between the different parts of the model. Colonising bacteria may originate anywhere within the system, allowing for all the major infection pathways, i.e. bacteria spreading up the outside surface or luminal surface, as well as an infection originating in the bladder (e.g. after a catheter is replaced). The catheter is treated as a rigid, open, cylindrical tube(with rotational symmetry). The bladder is assumed to contain a well-mixed pool of residual urine and the rate of urine production and the volume of residual urine are assumed to be constant. We assume that the catheter is colonised by a single bacterial species and base our parameters on *E. coli*, which is present in 40–70% of CAUTI[7]. Our model does not include the response of the human host and therefore does not distinguish between asymptomatic bacteriuria and symptomatic (CAUTI) infection[3]. Computer simulations of the

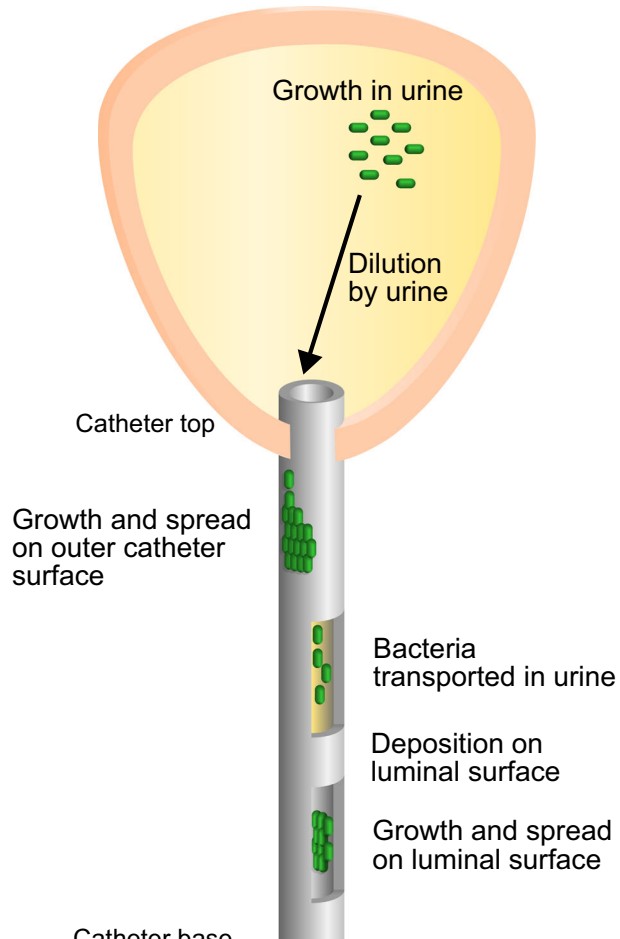

**Fig. 1 | Bacterial colonisation of a urinary catheter: schematic.** Bacteria grow and spread as a wave on the outside surface of the catheter. They then grow in the residual urine within the bladder, before being transported downwards by the flow within the catheter. Some bacteria attach to the inside of the catheter, where they grow and spread, eventually forming a biofilm.

Within the figure:
- Growth in urine
- Dilution by urine
- Catheter top
- Growth and spread on outer catheter surface
- Bacteria transported in urine
- Deposition on luminal surface
- Growth and spread on luminal surface
- Catheter base

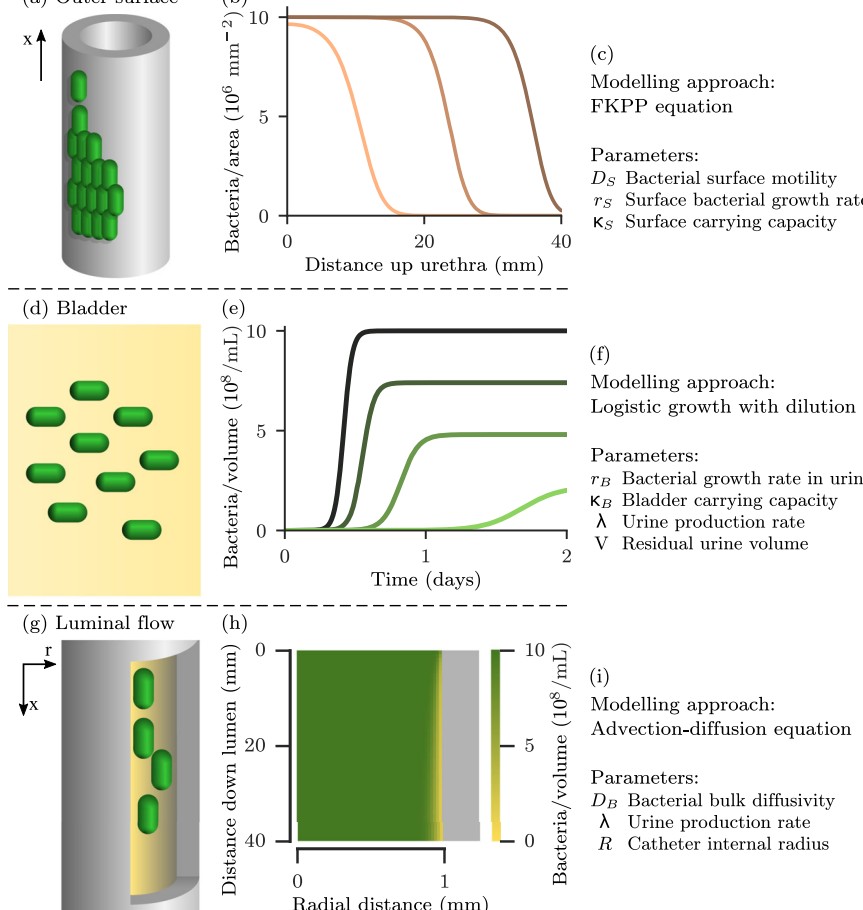

**Fig. 2 | Components of the mathematical model. a** Bacteria grow and spread as a wave on the outside surface of the catheter. **b** Behaviour of Eq.(1) ('Methods') subject to open boundaries and an initial inoculation of 100 bacteria per mm² at the bottom of the catheter($x = 0$), with all parameters as given in Table 1. Each curve is a snapshot of the surface density profile (bacterial abundance on the catheter surface at different distances along the catheter) at 24 (pink), 36 (light brown), and 48 (dark brown) h, respectively. **c** Physical parameters controlling outer surface model behaviour. **d** Bacteria grow in residual urine within the bladder. **e** Behaviour of Eq. (2), with an initial inoculation of $10^3$ mL$^{-1}$, subject to different dilution rates k$_D$. Each curve shows a computer simulation of the bacterial abundance (in cells per mL) in

the bladder, for dilution rates of 0 (darkest green), 0.36 (dark green), 0.72 (mid green), and 1.08 (light green) h$^{-1}$, respectively. **f** Physical parameters controlling bladder model behaviour. **g** Urine transports bacteria downwards through the catheter. **h** Numerical solution of Eqs. (3) and (4) gives the spatial distribution of bacterial abundance within the catheter. At the top of the catheter, bacteria are uniformly distributed within the urine with abundance $10^9$ mL$^{-1}$. However, further down the catheter, the urine close to the surface becomes depleted of bacteria, due to bacterial deposition on the luminal (inside) surface. **i** Physical parameters controlling luminal flow model behaviour.

model lead to predictions for the dynamics of the bacterial infection on the catheter and in the bladder, i.e. the abundance of bacteria in the urine and at different positions on the outside and luminal catheter surfaces, at different times. Here we briefly summarise the model components; full details of the model and its implementation, as well as background information, are given in 'Methods' and the Supplementary Information (sections I–III).

**Outside surface of the catheter.** Bacteria proliferate and spread on the outside surface of the catheter. We assume that if the infection initiates at the base of the catheter, it will spread up the catheter as a population wave (Fig. 2a–c). Detailed information on bacterial growth and proliferation on catheter surfaces are not currently available (see Supplementary Information I.G), therefore we adopt the simplest generic model that produces a population wave: the 1-dimensional Fisher-Kolmogorov-Petrovsky-Piskunov (FKPP) equation[29].

**Bladder.** Bacteria proliferate in the residual sump of urine that remains in the bladder during catheterisation[9,27,28], and then are removed from the bladder by urine flow. To model this, we combine a logistic growth

model[29] with a constant dilution rate that depends on the rate of urine production by the kidneys (Fig. 2d–f).

**Luminal flow.** Urine leaving the bladder flows through the catheter lumen, transporting bacteria downwards. Some of these bacteria adhere to the catheter surface, where they can form a biofilm. The bacterial density within the catheter (i.e. the abundance of bacteria within the urine at different positions in the catheter lumen) is modelled with an advection-diffusion equation (Fig. 2g–i), assuming a Poiseuille flow profile for the urine (Reynolds number ≈ 6). Attachment of bacteria to the luminal surface is incorporated as an absorbing boundary condition (see 'Methods' and Supplementary Information II.E).

**Luminal surface.** Bacteria that have been deposited on the luminal surface proliferate and spread. This is modelled by an FKPP equation with an additional source term due to the deposition of bacteria from the urine flow.

**Connecting the model parts.** We also account for movement of bacteria between the different parts of the model. The bladder and

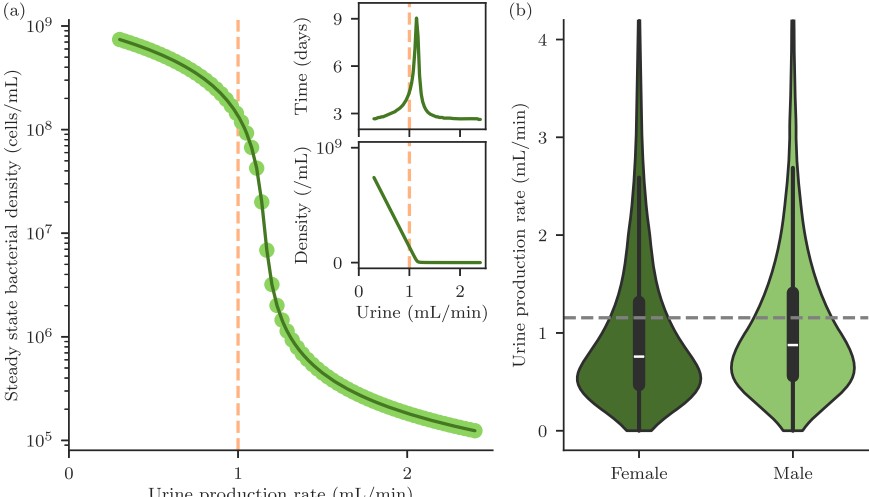

**Fig. 3 | Transition between bacteriuria and no bacteriuria, driven by urine production rate. a** As the rate at which the bladder contents are diluted by urine flow increases beyond the bacterial growth rate, there is a transition from a state with high bacterial abundance in the urine to a washed-out state with low bacterial abundance in the urine. Each point represents the long-time outcome of a computer simulation with a different value of the urine production rate. Lower inset shows the same data on a linear scale. Upper inset shows the time taken to attain the steady state, with a sharp peak at the transition. Also plotted(pink dashed) is the typical urine production rate for a patient, 1 mL/min[9]. **b** Violin plots showing the distribution of mean urine production rates in the US population. The box and whiskers show the quartiles, with the median in white. Female: a weighted sample ($n$ = 100,000) from a sample population of 11,011 adult females from ref. 32. Male: a weighted sample ($n$ = 100,000) from a sample population of 10,202 adult males from ref. 32. The grey dashed line represents the model estimation of the critical urine production rate for susceptibility to bacteriuria (i.e. the location of the transition in (**a**)), 1.16 mL/min. Patients with mean urine flow rates above this threshold are not predicted to develop bacteriuria, even if catheterised. Patients with mean flow rates below the threshold are predicted to be at greater risk of bacteriuria if catheterised.

outside surface are connected by detachment and attachment of bacteria to the top of the catheter, the outside and luminal surfaces are connected by bacterial migration across the top of the catheter, the bladder and luminal flow are connected by urine flow out of the bladder and the luminal flow and luminal surface are connected by bacterial deposition on the surface (see 'Methods', and Supplementary Information II.C, Fig. S4).

## Long-term susceptibility to bacteriuria depends on urine production rate

Simulating our model for long times (several months) allows us to predict how the infection risks of long-term catheterisation depend on factors relating to the patient, the catheter, and the infecting bacterial strain. For simulations longer than a few weeks, the bacterial abundances no longer change in time, i.e. the model reaches a steady state. We observe two possible outcomes. In one outcome, which we associate with bacteriuria, the abundance of bacteria in the urine is high, while in the other outcome, which we associate with no bacteriuria, the abundance of bacteria in the urine is low. Which of these outcomes happens depends critically on the rate of urine production by the kidneys, which is correlated with the fluid intake of the patient. Varying the urine production rate leads to a transition in the model's behaviour (Fig. 3a). For low urine production rates, bacteria grow fast enough in the bladder to overcome dilution and they become abundant in the urine. In contrast, for high urine production rates, the dilution rate exceeds the maximal rate at which the bacteria can grow in the bladder, so that the bacterial abundance in the urine falls dramatically. From a mathematical perspective, this transition is continuous in the first order with an initial near-linear dependence of the steady state bacterial population density on the urine production rate(Fig. 3a lower inset); however the time taken to attain the steady state diverges at the transition (Fig. 3a upper inset), which is characteristic of a continuous phase transition[30] and suggestive of an ecological tipping point[31] (for further mathematical details see Supplementary Information IV.A).

A typical value for the urine production rate in humans is 1 mL/min (dashed line in Fig. 3a)[9]. This value lies very close to the threshold of 1.16 mL/min for the transition between bacteriuria and no bacteriuria that is predicted by our model (Fig. 3a), suggesting that individual long-term catheterised patients might show very different levels of bacteriuria depending on their fluid intake. The Centres for Disease Control and Prevention (CDC) provides data on urine production rates for randomly sampled adults in the US population[32] (Fig. 3b). Comparing this data with the threshold urine production rate for long-term bacteriuria that is predicted by our model, we predict that 70.0 ± 0.8% of females and 65.4 ± 0.7% of males are susceptible to catheter-associated bacteriuria, i.e. they are at risk of bacteriuria if they undergo long-term catheterisation because they have urine production rates below the bacteriuria-no bacteriuria threshold. The remaining fraction of the population has urine production rates above the threshold and are predicted not to be susceptible to catheter-associated bacteriuria(for model parameters see Table 1; for further discussion of the CDC data analysis, see Supplementary Information IV.B). Our model therefore predicts that increasing urine production rate, e.g. by increasing fluid intake, decreases an individual's risk of developing bacteriuria. For example, we predict that increasing urine production rate by 300 mL/day (0.21 mL/min) would result in a decrease in the relative risk of bacteriuria of 8.9 ± 0.3% (females) or 10.5 ± 0.4% (males) (see Supplementary Information IV.B, Table S1). This analysis might explain why some patients suffer recurrent CAUTI infections while others do not[18] and supports the concept of increasing fluid intake as an effective measure to decrease bacteriuria and thus decrease progression to CAUTI in long-term catheterised patients.

The urine production rate is a key parameter in our model, since it controls the rate at which bacteria are removed from the bladder in the urine flow; this corresponds to a dilution term in the mathematical model with rate $k_D$ (see Eq. (2) in 'Methods'). The urine production rate also controls the rate of urine flow through the catheter lumen(denoted by $\lambda$; see Eq.(4) in 'Methods'), which in turn determines the rate at which bacteria are deposited on the luminal surface. Other

**Table 1 | Model parameters**

| Parameter | | Default value in simulations | Justification and expected range |
|---|---|---|---|
| Urethral length | $L$ | 40 mm | Women: 40 mm. Men: 160 mm[9] |
| Residual urine volume | $V$ | 50 ml = $5 \times 10^4$ mm$^3$ | 10–100 ml[9], may be up to 500 ml in immobile patients[28] |
| Urine production rate | $\lambda$ | 1 ml min$^{-1}$ = 16.7 mm$^3$s$^{-1}$ | 1 ml min$^{-1}$ [9] |
| Dilution rate | $k_D$ | 1.2 h$^{-1}$ = $3.33 \times 10^{-4}$ s$^{-1}$ | $k_D = \lambda/V$ |
| Catheter internal radius | $R$ | 1 mm | Outer diameter: 4–5 mm[9], wall thickness is manufacturer dependent |
| Bacterial surface diffusivity | $D_S$ | $10^2$ µm$^2$s$^{-1}$ = $10^{-4}$ mm$^2$ s$^{-1}$ | Assume surface motility equal to bulk diffusivity: $D_S = D_B$ |
| Bacterial bulk diffusivity | $D_B$ | $10^2$ µm$^2$ s$^{-1}$ = $10^{-4}$ mm$^2$ s$^{-1}$ | 100 µm$^2$ s$^{-1}$ for *E. coli*[63] |
| Catheter surface bacterial growth rate | $r_S$ | 0.69 h$^{-1}$ = $1.93 \times 10^{-4}$ s$^{-1}$ | Doubling time of *E. coli* on poor media is ~1 h[64] |
| Bacterial growth rate in bladder | $r_B$ | 1.39 h$^{-1}$ = $3.85 \times 10^{-4}$ s$^{-1}$ | Doubling time of *E. coli* in UTIs is 30–35 min[65] |
| Catheter surface carrying capacity | $\kappa_S$ | $10^9$ cm$^{-2}$ = $10^7$ mm$^{-2}$ | Biofilms of $5 \times 10^9$ cm$^{-2}$ have been found on catheters[57] |
| Bladder carrying capacity | $\kappa_B$ | $10^9$ ml$^{-1}$ = $10^6$ mm$^{-3}$ | *E. coli* in mice[65]: $10^8$ CFUg$^{-1}$ and in LB[66]: $10^9$ cells mL$^{-1}$ |
| Bacterial detachment rate | $k_d$ | $1.93 \times 10^{-4}$ s$^{-1}$ | Assume that all new growth detaches: $k_d = r_S$ |
| Bacterial attachment rate | $k_a$ | $1.26 \times 10^{-6}$ s$^{-1}$ | Smoluchowski diffusion rate limited constant[67]: $4\pi D_B \cdot 1$ µm |

The values used in the model are listed together with the expected range of values for each parameter.

parameters of the model also influence $k_D$; these are the growth rate of bacteria in the urine and the volume of residual urine in the bladder (Supplementary Information II.A and Fig. S2). These parameters are also predicted to have a strong influence on the susceptibility of long-term catheterised patients to bacteriuria. This is consistent with a previous (uncatheterised) micturition model, which pointed to the relevance of both urine production rate and the bacterial growth rate in the bladder for urine infection dynamics[26]. Therefore, measures to decrease the volume of residual urine in the catheterised bladder could also be effective in preventing CAUTI in long-term catheterised patients.

The bacteriuria transition predicted by our model is similar to the washout phenomenon that occurs in continuous bacterial culture, i.e. in a chemostat, when the dilution rate approaches the bacterial maximal growth rate[33,34] so that the bacteria in the chemostat device cannot grow fast enough to keep up with dilution and are washed out of the device. Indeed, when plotted on a linear scale (Fig. 3a lower inset), the transition in the catheter model appears indistinguishable from washout, with the bacterial abundance in the urine approaching zero as the urine production rate approaches the bacterial growth rate. However, in the logarithmic plot (Fig. 3a main), we observe a tail caused by re-population of the bladder from bacteria that have adhered to the catheter surfaces. Hence, the presence of the catheter allows an infection to persist even if the urine production rate is high enough to wash it out of the bladder, since the infection can be re-seeded from the biofilm on the catheter surface (see Supplemental Material IV.C for further discussion). Indeed, catheter-associated biofilms are known to act as a bacterial reservoir, leading to persistent re-colonisation of the bladder[35]. Therefore, our model predicts that even patients who do not experience bacteriuria (e.g. because they have a high urine production rate; Fig. 3b) may still experience adverse outcomes such as catheter blockage.

## Timing of bacteriuria depends on speed of spread of bacteria along catheter

Even though bacteriuria is an almost inevitable consequence of long-term catheterisation[6], patients who are catheterised for short times may avoid bacteriuria if the catheter is removed early enough. This is consistent with the fact that duration of catheterisation is the biggest risk factor for developing CAUTI[4,5,17].

Our model suggests that for short-term catheterised patients, the clinical outcome depends on whether the catheter is removed before the bacteria have reached the bladder and multiplied to cause bacteriuria. To assess the short-term outcome, we track in our computer

simulations the time after infection at which bacterial abundance in the urine reaches a threshold, which we denote as the detection threshold for bacteriuria (Fig. 4). Here we assume that colonisation is initiated by bacteria on the outside of the catheter, where the urethra meets the skin, and that bacteria migrate up the outside surface of the catheter to the bladder (see 'Methods').

In our model, the factors that control the outcome for short-term catheterisation are very different from those that are relevant for long-term catheterised patients. In particular, the time to detection of bacteriuria is almost independent of the urine production rate for the parameters of Table 1 (Fig. 4a). However, we find that the length of the catheter, which corresponds to the length of the urethra, plays a crucial role: the time to detection of bacteriuria depends linearly on the catheter length (Fig. 4b). The main factor determining urethral length in humans is sex, with a typical urethral length for a woman being 40 mm, and for a man 160 mm[9]. This result is therefore consistent with the fact that sex is a well-established risk factor for CAUTI[5,17].

In the model, the time to bacteriuria is controlled by how long the bacterial wave takes to spread up the catheter to reach the bladder. Therefore, as well as catheter length, the speed of the bacterial wave is important, which in our model depends on the bacterial growth rate and mobility on the catheter surface (Eq. 1). Antimicrobial coated catheters, in which the surface is coated or impregnated with an antimicrobial substance such as silver ions or nitrofural, are commercially available, although they have shown mixed efficacy in preventing CAUTI in clinical trials[23,24]. These coatings aim to alter the interaction between bacteria and the catheter surface by killing bacteria or preventing adhesion. Our model suggests that an antimicrobial coating would not affect the long-time clinical outcome (bacteriuria or no bacteriuria) for long-term catheterised patients, but it could delay the onset of bacteriuria in short-term catheterised patients, especially male patients (who have a longer urethra than female patients), by altering the time taken for the bacterial wave to reach the bladder.

In our computer simulations, we assumed that migration of bacteria up the catheter is slow relative to their proliferation in the bladder (Table 1). Different outcomes would be expected if bacteria are either highly motile on the catheter surface or very slow-growing in the bladder (see Table 2). In this fast-migration/slow-growth regime, the rate-limiting step would be growth in the bladder rather than migration up the catheter and parameters associated with bacterial growth in the bladder (urine production rate, residual urine volume and bacterial growth rate in the bladder) would become relevant for the short-term colonisation dynamics. This regime is discussed in more detail in the Supplementary Information (IV.D).

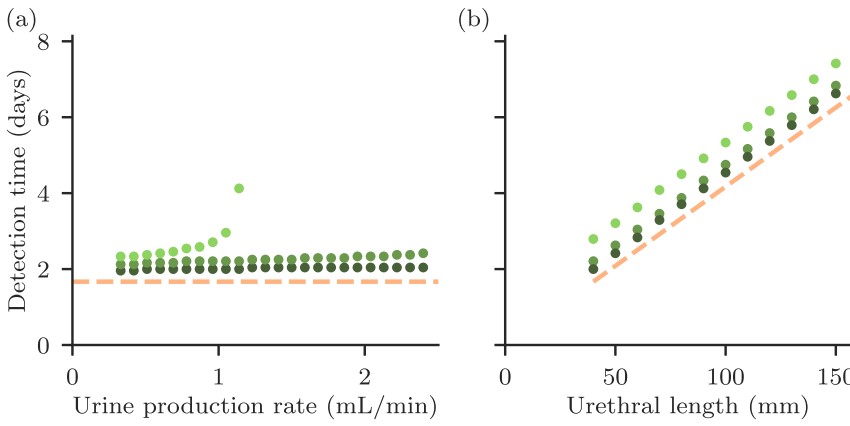

**Fig. 4 | Time to detection of bacteriuria.** The detection time is defined as the time until the bacterial abundance in the urine exceeds a detection threshold of $10^7$ (light green), $10^5$ (mid green), or $10^3$ (dark green) cells/mL, mimicking tests of different sensitivity. Also plotted (pink dashed) is the theoretically predicted time for the FKPP wave to travel the length of the catheter $L/2\sqrt{r_S D_S}$. **a** Time to detection of bacteriuria for different values of the urine production rate. For the high-sensitivity test (threshold $10^3\,\mathrm{mL}^{-1}$), bacteriuria is eventually detected, no matter how high the urine production rate, because even in the washout regime, the presence of the catheter ensures that there are some bacteria in the urine. In contrast, low sensitivity tests do not detect bacteriuria when the model is run for high urine

production rates, because the bacterial abundance in the urine is very low in the no bacteriuria regime (Fig. 3a). For CAUTI diagnosis, clinical guidelines[5] require both clinical symptoms and bacterial counts $> 10^3$ colony forming units (CFU) $\mathrm{mL}^{-1}$ while a diagnosis of significant bacteriuria requires a higher bacterial count $> 10^5$ CFU $\mathrm{mL}^{-1}$. **b** Time to detection of bacteriuria for different values of the urethral length. The time for bacteriuria to develop increases linearly with the urethral length. The vertical axis scale is the same as in (**a**). Note that in this case, there is no significant difference between different testing sensitivities, as the urethral length dominates the timescale. All model parameters are as given in Table 1 (for comparison with Fig. 3, note that the urine production rate is 1 mL/min).

## Different sources of infection produce different patterns of biofilm coverage

Most CAUTI are thought to originate on the outside of the catheter, for example, from the gastrointestinal tract via the skin of the meatus and perineum[20–22]. However, bacteria can also contaminate the drainage bag or port and ascend the luminal surface of the catheter[22]. A third pathway is contamination during catheter insertion, which is thought to account for around 5% of CAUTI[7]. Sources within the bladder are also possible, such as a urinary tract infection prior to catheterisation, while in mouse models, persistent intracellular bacterial communities in the epithelial cells that line the bladder can also act as an infection source[21,36,37]. In our model, different sources of infection lead to different spatial patterns of bacterial abundance on the outside and luminal surfaces of the catheter during colonisation (Fig. 5), although the model always predicts eventual complete coverage of both surfaces by bacteria.

If bacterial colonisation originates on the outside at the catheter base (i.e. from the skin), a wave of bacteria spreads up the outside surface. At early times, therefore, the model predicts high bacteria coverage on the lower part of the outside surface only, while the upper part of the outside surface and the luminal surface are still uncolonised (Fig. 5a). If the origin is instead at the base of the luminal surface (e.g. from the drainage bag), the bacterial wave spreads up the inner surface, so that at early times bacterial coverage is high only on the lower part of the luminal surface (Fig. 5b). If the colonisation originates in the bladder, the entire luminal surface rapidly becomes colonised and a bacterial wave also propagates down the outside surface from the top (Fig. 5c). Finally, if the catheter were to become contaminated on insertion such that bacteria become spread over the entire outside surface, the model predicts rapid bacterial growth across the entire surface (Fig. 5d). This would imply a breach of the sterile conditions under which clinical catheter insertion should be performed.

Inspection of bacterial abundance on the outside and luminal surfaces of catheters removed at an early stage is, in principle, possible, and would provide a useful test for hypotheses and models of catheter colonisation. Such measurements are not routinely performed, but strikingly similar patterns to those predicted by our model have been observed for bacterial colonisation of an in vitro model of a

catheterised bladder[38,39]. Fitting our model to this data allows us to extract parameters for bacterial growth rate and motility on the catheter surface (see Supplementary Information, IV.E). In addition, this approach could be helpful for optimising catheter management protocols. For example, if a catheter has been removed due to infection, it could be inspected for bacterial colonisation of the lower and upper outside vs luminal surfaces, potentially informing about the most likely source of infection and informing targeted infection prevention measures for further catheterisation.

## Discussion

Urinary catheters are notoriously prone to colonisation by bacteria, with high societal and economic costs, yet the pathways by which colonisation occurs remain unclear and interventions to mitigate infection have shown mixed results[9,23,27]. In this work, we present a quantitative mathematical model for the dynamics of urinary catheter colonisation that integrates bacterial population dynamics on the catheter surfaces and in the bladder with urine flow. Our model suggests explanations for the longstanding observations that some patients experience frequent infections while others do not[18], some infections develop more rapidly than others[19], sex and duration of catheterisation are key risk factors for CAUTI[6,40], and antimicrobial coatings show mixed efficacy[23,24].

Our model allows us to investigate which factors relating to the patient, bacterial strain, and catheter design influence clinically relevant outcomes for long-term and short-term catheterisation scenarios. These factors are embodied as model parameters. The most important finding of our study is that clinical outcomes for long-term and short-term catheterisation are controlled by different factors (as summarised in Table 2). Therefore, qualitatively different types of intervention may be needed to prevent infection for long-term vs short-term catheterised patients.

Long-term catheterisation is mimicked by simulating our model over several months, by which time the catheter is fully colonised and the model's behaviour no longer changes (it reaches a steady state). Here, we observe two distinct outcomes corresponding to abundant and non-abundant bacteria in the urine, which we term bacteriuria and no bacteriuria. The urine production rate emerges as a key factor that

**Table 2 | Factors controlling short-term and long-term outcomes in the model**

| Parameter | | Short-term outcome | Long-term outcome |
|---|---|---|---|
| Urethral length | $L$ | ✓ | |
| Residual urine volume | $V$ | fast migration/slow growth | ✓ |
| Urine production rate | $\lambda$ | fast migration/slow growth | ✓ |
| Catheter internal radius | $R$ | | |
| Bacterial surface diffusivity | $D_S$ | ✓ | |
| Bacterial bulk diffusivity | $D_B$ | | |
| Catheter surface bacterial growth rate | $r_S$ | ✓ | |
| Bacterial growth rate in bladder | $r_B$ | fast migration/slow growth | ✓ |
| Catheter surface carrying capacity | $\kappa_S$ | | |
| Bladder carrying capacity | $\kappa_B$ | | |

Model parameters are classified into those affect the short-term and long-term clinical outcome. Importance of parameters is quantified by a global sensitivity analysis (see Supplementary Information IV.F). For the short-term outcome, some parameters are relevant only if bacteria are highly motile on the catheter surface or very slow-growing in the bladder (see Supplementary Information IV.D for a detailed discussion of how the model behaves differently depending on the bacterial motility on the catheter surface). This corresponds to $\alpha \lesssim 1$, as defined by Eq. S28 of the Supplementary Information, and is indicated in the table by 'fast migration/slow growth'.

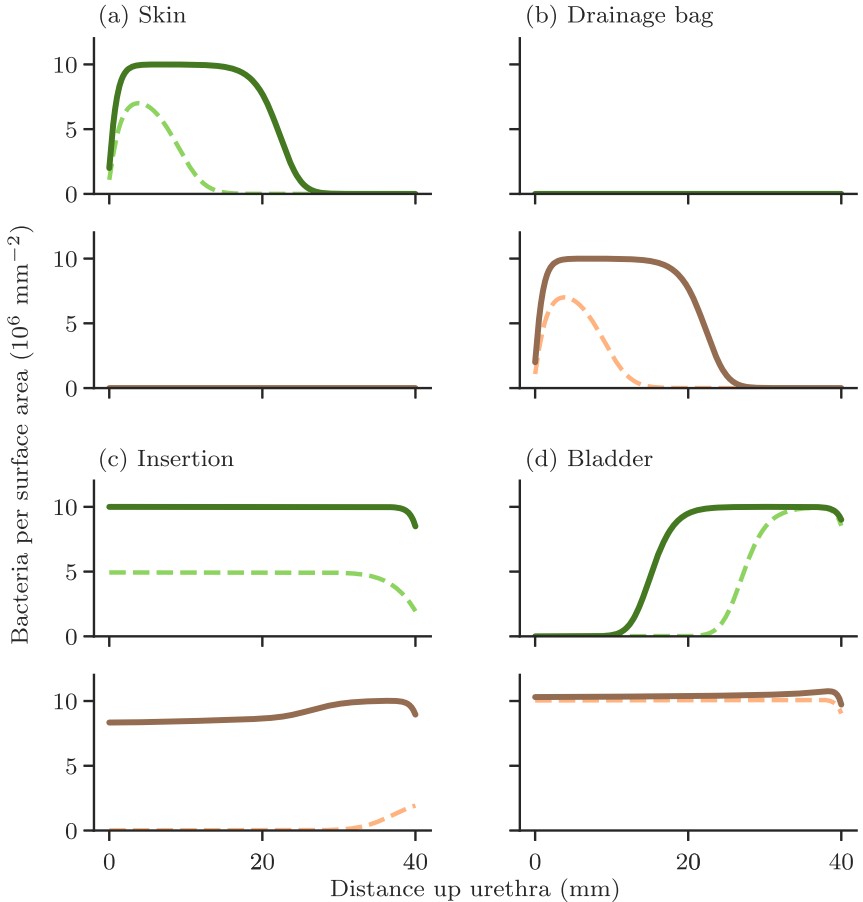

**Fig. 5 | Patterns of bacterial colonisation along the catheter differ for different infection scenarios.** The predicted bacterial abundance at different positions along the outside and luminal catheter surfaces is shown for timepoints 24 and 36 h after infection, for 4 different infection scenarios. For each infection scenario, predictions for the outside surface are given in the upper panel, in green, and for the luminal surface in the lower panel, in pink. Predictions for 24 h are shown as dashed lines while predictions for 36 h are shown as solid lines. Here, a distance of 0 mm corresponds to the part of the catheter that contacts the skin or drainage bag, and 40 mm corresponds to the part that contacts the bladder. All model parameters are as given in Table 1. The four infection scenarios that are modelled are: **a** The skin acts as a reservoir from which infection can spread. **b** The drainage bag becomes contaminated, and bacteria ascend the catheter lumen. **c** The catheter is inoculated uniformly at time of insertion. **d** Bacteria are already present in the bladder prior to insertion. Note that here we assume the catheter is the same length as the urethra; in reality, there is an additional length of catheter beyond the urethra that connects to the drainage bag, so we would expect infections ascending from the drainage bag to have somewhat longer establishment times than shown in **b**.

controls a transition between bacteriuria and no bacteriuria. Other parameters associated with the bladder—residual urine volume and bacterial growth rate in urine—are also relevant for this transition. Comparing the measured distribution of human urine production rates with our predicted threshold value for the bacteriuria—no bacteriuria transition suggests that ~70% of the population is susceptible to catheter-associated bacteriuria, potentially explaining observed differences in susceptibility to CAUTI among individual patients. The model also suggests that individuals might reduce their susceptibility by increasing fluid intake, hence increasing the rate of removal of bacteria from the bladder above the bacteriuria threshold. In fact, drinking more water is known to be protective against urinary tract infections[26,41–43]. In non-catheterised female patients with recurrent UTI, increasing water intake by 1.5 L per day was found to almost halve the mean number of cystitis incidents over 12 months[41].

For short-term catheterisation, the early-time colonisation dynamics of the model are relevant, i.e. how fast bacteria migrate up the catheter and colonise the bladder. A short-term catheterised patient will only develop bacteriuria (and potentially CAUTI) if the onset time for bacteriuria is shorter than the duration of catheterisation. Hence, delaying the onset of bacteriuria can be a useful strategy for short-term catheterised patients. Our model shows that the time taken by bacteria to migrate up the catheter depends linearly on urethral length, consistent with the fact that sex is a strong risk factor[40,44,45]. Other parameters associated with the bacteria-surface interaction—bacterial growth rate and motility on the surface—are also relevant for the onset time. Thus, our model suggests that interventions that alter bacteria-surface interactions, such as antimicrobial coatings, might be effective for short-term catheterised patients only.

Reports on the efficacy of antimicrobial coatings have been mixed[23,24]. A systematic review in 2008 found that both silver-alloy coated and antibiotic-impregnated catheters were effective in reducing the risk of catheter-associated urinary tract infection for short-term catheterised patients (in that study, defined as <1 week) but were not effective for long-term catheterised patients[23]. However, a subsequent randomised control trial in 2012 that focused on short-term catheterisation (here defined as <2 weeks) found that silver-alloy coated catheters were not effective, while antibiotic-impregnated catheters had only limited efficacy[24]. The observed loss of efficacy for long-term catheterised patients is consistent with our model findings. Moreover, our model might also be able to explain the discrepancy between the two reports. A key finding of our model is that timescales are critical for clinical outcomes, so the different outcomes of the two reports might be explained by the different definitions of 'short-term': 1 week[23] vs 2 weeks[24]. Furthermore, our model predicts that antimicrobial coatings are likely to have significantly higher efficacy in males than females, since urethral length is a controlling parameter for short-term outcomes. Neither study[23,24] distinguished efficacy in males vs females, but we note that the 2008 review[23] (which found positive short-term efficacy) included more males than females, while 62% of the participants of the 2012 trial[24] (which found no efficacy) were female. We suggest that it is important for future studies of intervention efficacy to differentiate between males and females, as well as to standardise definitions of short-term versus long-term catheterisation duration.

CAUTI is defined as a symptomatic infection involving pain or fever that should be treated by catheter removal and possibly antibiotics; in contrast, asymptomatic bacteriuria may not need to be treated[5,7]. Since our model does not describe the immune response to bacterial infection, it cannot predict the presence or absence of pain or fever symptoms. However, it does make a clear distinction between bacteriuria and bacterial colonisation of the catheter lumen (i.e. biofilm formation that can lead to catheter blockage). In the model, bacteriuria happens earlier than biofilm formation in the lumen, suggesting that detection of bacteriuria does not necessarily imply that biofilm is present in the lumen. Interestingly, the model also predicts that a catheter could block without bacteriuria being detected at all. This happens in the regime of high urine flow, where the bacterial abundance in the urine is low, but the lumen can still become colonised.

Intriguingly, our model also suggests that the spatial patterns of biofilm on infected catheters could provide information on the source of infection. Observations in the literature to date are limited, but appear similar to our model predictions[38,39,46], suggesting that closer observation of the patterns of biofilm on infected catheters could be highly informative, both for model testing and, potentially, to inform catheter management protocols.

In our model, differences in clinical outcome among different patients are assumed to arise because of variability in the parameters that represent characteristics of the patients (e.g. urethral length and urine production rate) and the infecting bacterial strains (e.g. growth and motility parameters). Variable outcomes might also arise due to inherent stochasticity, e.g. in bacterial growth and motility dynamics[33,47] or the initial arrival of bacteria on the catheter[48]. This inherent stochasticity is not included in our model so far. The variability in patient and strain parameters is already large (see, e.g. Fig. 3b for urine production rate); it would be interesting in future to investigate the magnitude of the additional contributions arising from inherent stochasticity.

Our model could be extended to bring it closer to clinical reality. To account for the consequences of immune activation on bacterial dynamics and host response, one should combine our model with existing immune system models[49]—this would allow, for example, predictions of fever symptoms and differences in CAUTI progression for immunosuppressed patients[15]. It would also be interesting to model biofilm growth on the catheter surfaces in more detail, including, for example, the role of quorum sensing in biofilm initiation[48,50], triggered biofilm dispersal[51] and the interplay between biofilm structure and fluid flow in the catheter lumen[52]. It may be especially interesting to investigate how bacteria that disperse from biofilm (either as single cells or aggregates) on one part of the catheter surface could seed new biofilms on other parts of the catheter, potentially accelerating catheter blockage. It would also be interesting to investigate how a developing biofilm in the lumen alters the urine flow pattern, in turn altering the patterns of bacterial deposition on the lumen surface and potentially accelerating catheter blockage. However, to properly model catheter blockage, one should also include not only the possibility of crystalline biofilm formation due to alkalinisation of the urine by *Proteus mirabilis*[21,53] but also the possible role of debris from dead bacteria, sloughed epithelial cells and/or blood clots. We have also ignored the possibility of biofilm growth on the catheter balloon, whose different material properties might affect biofilm formation and growth dynamics. Our modelling of bacterial dynamics in the bladder could also be extended to account for patient-to-patient differences in nutrient availability, most obviously the increased glucose concentration in diabetic patients, which may be linked to the observed increased risk of urinary tract infection[54,55]. Urine is a complex medium for which detailed bacterial growth models are lacking; development of such models would be a useful direction for future research.

The Foley catheter has changed little in design since its invention in the 1930s, even though it is highly prone to infection[9,27]. The introduction of sealed drainage bags in the 1960s reduced the chance of infection[56,57] and, more recently, meatal cleaning with chlorhexidine before catheter insertion has been found to reduce bacteriuria and CAUTI incidence[58]; however, the use of various antimicrobial catheter coatings has met with mixed success[23,24]. It is a major challenge to design effective and practical interventions to reduce CAUTI, but quantitative mathematical models could help understand the

mechanisms by which catheter colonisation occurs and hence guide a more nuanced approach to the targeted design of intervention strategies.

## Methods

### Model definition

**Outside surface.** Bacterial growth and spreading on the outside surface of the catheter is modelled in 1-dimension, the distance $x$ up the catheter (Fig. 2a), using an FKPP equation[29]:

$$\frac{\partial n}{\partial t} = D_S \frac{\partial^2 n}{\partial x^2} + n r_S \left(1 - \frac{n}{\kappa_S}\right). \tag{1}$$

Here, $n(x, t)$ is the bacterial surface density at point $x$ at time $t$, $D_S$ the diffusivity of the bacteria on the catheter surface, $r_S$ the bacterial growth rate on the catheter surface, and $\kappa_S$ the bacterial carrying capacity of the catheter surface (see Table 1).

**Bladder.** Assuming the residual urine in the catheterised bladder[9,27,28] to be well-mixed and of constant volume, we model bacterial dynamics with a logistic growth equation, supplemented by a dilution term[29]:

$$\frac{d\rho}{dt} = r_B \rho \left(1 - \frac{\rho}{\kappa_B}\right) - k_D \rho. \tag{2}$$

Here, $\rho(t)$ is the bacterial volume density at time $t$, $r_B$ is the (maximum) growth rate of the bacteria in urine, and $\kappa_B$ is the carrying capacity of urine. The dilution rate is given by $k_D = \frac{\lambda}{V}$, where $\lambda$ is the rate of urine production by the kidneys, and $V$ is the volume of residual urine in the bladder (see Table 1 and Supplementary Information II.A, Fig. S1).

**Luminal flow.** Transport of bacteria in the urine that flows through the catheter lumen is modelled with a 2-dimensional advection-diffusion equation[59,60], in the radial distance $r$ and the longitudinal distance down the catheter $x$ (geometry illustrated in Fig. 2c):

$$\frac{\partial \sigma}{\partial t} = D_B \left(\frac{\partial^2 \sigma}{\partial r^2} + \frac{1}{r}\frac{\partial \sigma}{\partial r}\right) - u(r)\frac{\partial \sigma}{\partial x} \tag{3}$$

$$u(r) = \frac{2\lambda}{\pi R^4}\left(R^2 - r^2\right). \tag{4}$$

Here, $\sigma(r, x, t)$ is the bacterial volume density in the urine, $D_B$ is the diffusivity of bacteria within urine, and $u(r)$ is the flow profile of the urine, with $R$ being the internal radius of the catheter, and $\lambda$ the rate of urine production. A Poiseuille flow profile is expected, since the Reynolds number is $\approx 6$ (see Supplementary Information II.B and Fig. S3 for further discussion).

**Luminal surface.** Bacterial growth and spreading on the luminal surface is modelled with a 1-dimensional FKPP equation, supplemented with a source term $j(x)$ accounting for deposition of bacteria from the urine (see Eq. (11) below):

$$\frac{\partial m}{\partial t} = D_S \frac{\partial^2 m}{\partial x^2} + r_S m\left(1 - \frac{m}{\kappa_S}\right) + j(x). \tag{5}$$

Here, $m(x, t)$ is the bacterial surface density; $D_S$, $r_S$ and $\kappa_S$ are the diffusivity, growth rate, and carrying capacity (as above; see also Supplementary Information II.B and Fig. S3).

**Connecting the outside surface and bladder.** Bacterial transfer from the top of the outside surface to the bladder is incorporated by adding the following term to the right-hand side of Eq. (2):

$$\frac{k_d S_c n(0, t) - k_a V_c \rho(t)}{V}, \tag{6}$$

where $V$ is the volume of residual urine in the bladder, $n(0, t)$ is the bacterial surface density at the top of the catheter, $S_c$ is the area of catheter surface in contact with the bladder, $V_c$ is the volume of urine surrounding the catheter into which bacteria can transfer, and $k_d$ and $k_a$ are respectively the rates at which bacteria detach from and attach to the catheter surface. Bacterial transfer in the other direction, from the bladder to the outside surface, is incorporated by a corresponding surface density flux at the top of the catheter, which is added to the right-hand side of Eq. (1):

$$\frac{k_a V_c \rho(t) - k_d S_c n(0, t)}{S_c}. \tag{7}$$

These coupling terms are derived in the Supplementary Information II.D; see also Fig. S4.

**Connecting the outside and luminal surfaces.** Since we do not model the catheter eyelets, we assume that bacteria can spread freely across the top of the catheter between the outside and luminal surfaces. This is represented by a continuity condition on the bacterial density, connecting Eqs. (1 and 5):

$$n(x = 0, t) = m(x = 0, t),$$
$$\left.\frac{\partial n}{\partial x}\right|_{x=0} = \left.\frac{\partial m}{\partial x}\right|_{x=0}. \tag{8}$$

**Connecting the bladder and luminal flow.** To ensure continuity of bacterial density in the urine leaving the bladder and entering the lumen, we impose the following coupling between Eqs. (2 and 3):

$$\sigma(r, x = 0, t) = \rho(t). \tag{9}$$

**Connecting the luminal flow and luminal surface.** Deposition of bacteria from the urine onto the luminal surface is incorporated via an absorbing boundary condition for Eq. (3), i.e. the catheter wall is assumed to be a perfect sink. Deposition onto the surface depletes the urine of bacteria close to the surface (Fig. 2c). The flux of deposited bacteria appears as the source term $j(x)$ for the luminal surface in Eq. (5). To calculate this flux, we compute the rate at which bacteria hit the boundary $r = R$ for the longitudinal distance $x$ down the catheter:

$$j(x) = -D_B \left.\frac{\partial \sigma}{\partial r}\right|_{r=R}. \tag{10}$$

This flux can be obtained either by numerical solution of Eqs. (3 and 4), or by an analytic approximation[61], which gives

$$j(x) = 0.5835 D_B \rho \sqrt[3]{\frac{\lambda}{R^3 D_B}} \frac{1}{\sqrt[3]{x}}. \tag{11}$$

Details of the analytical approximation leading to Eq. (11) can be found in the Supplementary Information II.E.

### Model implementation

The model was simulated numerically using a forward-time centred-space method for the outside and luminal surface and a forward Euler method for the bladder. An approximate analytical solution for the bacterial flux onto the luminal surface was used to reduce the computational requirements (see Supplementary Information III).

## Parameters

Table 1 lists the parameters used in our model, their values and their expected range in a clinical setting.

## Data analysis

Analysis of the data produced by the numerical model was performed in Python 3.11.9 using matplotlib 3.8.4, numpy 1.26.4, pandas 2.2.2, and seaborn 0.13.2. The code used to produce all figures is available (see Code Availability).

## Statistical analysis

Data on urine production rates in the US population were obtained from the CDC National Center for Health Statistics(NCHS) National Health and Nutrition Examination Survey(NHANES)[32] and analysed using Stata 18.5. The dataset was constructed by combining four data cycles—2009–2010, 2011–2012, 2013–2014, and 2015–2016—with weightings adjusted accordingly as 1/4× wtmec2yr. For each individual, the urine production rate was calculated as the mean of the recorded urine flow rates, and the individuals with no recorded urine flow rates were removed. Subpopulation analysis was performed for males aged > 18 years($n = 10,202$) and females aged > 18 years($n = 11,011$). No outliers were removed during statistical analysis; however, in Fig. 3b the upper tail of the distribution has been cropped at 4 mL/min for ease of viewing (194 males and 154 females in the survey population have urine production rate > 4 mL/min).

## Sensitivity analysis

A global sensitivity analysis (Supplementary Material IV.F) was performed in Python 3.11.9 using SALib 1.5.1, with $n = 360,448$ model runs.

## Reporting summary

Further information on research design is available in the Nature Portfolio Reporting Summary linked to this article.

# Data availability

All datasets generated and analysed during the current study are available in the Zenodo repository with the identifier https://doi.org/10.5281/zenodo.15001619[62].

# Code availability

The computational model used in the current study is available in the Zenodo repository with the identifier https://doi.org/10.5281/zenodo.15001619[62].

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

## Acknowledgements

We gratefully acknowledge our colleague Professor Fiona Denison (1970–2022), who motivated and inspired this work. We also acknowledge discussions with Michael Bauer, Aidan Brown, Craig Breheny, Anne Busch, Fiona Denison, Ashutosh Deshpande, Susana Direito, Paul Flowers, Adam Gow, Voula Granitsiotis, Bettina Löffler, Gail McConnell, Tim Nuttall, Freya Oswald, Liam Rooney, Sally Stewart, Alice Street, Bartlomiej Waclaw, Ellen Young and the NHS Lothian Bladder and Bowel Nursing Team. This work was supported by EPSRC via the Scottish Doctoral Training Centre in Condensed Matter Physics (grant number EP/L015110/1), by the European Research Council under Consolidator Grant 682237 EVOSTRUC, by the Scottish Universities Life Science Alliance via an AMR Seed Award, and by the Excellence Cluster Balance of the Microverse (EXC 2051—Project-ID 390713860) funded by the Deutsche Forschungsgemeinschaft (DFG). The authors acknowledge the use of the UCL Myriad High Performance Computing Facility (Myriad@ UCL) and associated support services in the completion of this work. For the purpose of open access, the author has applied a Creative Commons Attribution (CC BY) licence to any Author Accepted Manuscript version arising from this submission.

## Author contributions

F.B., N.B., R.J.A. and S.T. conceived the project. F.B. and R.J.A. developed the mathematical model. F.B. developed the computational model, performed simulations, and analysed results. M.P. and N.B. contributed

clinical interpretation. C.A.B., R.J.A. and S.T. supervised the project. C.A.B., F.B. and R.J.A. wrote the manuscript. M.P. edited the manuscript. All authors discussed the results and commented on the manuscript.

## Funding

## Competing interests
The authors declare no competing interests.
