## [Transparent Peer Review file · Nature Communications]

Different factors control long-term versus short-term outcomes for bacterial colonisation of a urinary catheter

Corresponding Author: Professor Rosalind Allen

Version 0:

Reviewer comments:

Reviewer #1

(Remarks to the Author)

Review of: Different factors control long-term versus short-term outcomes for bacterial colonisation of a urinary catheter

Overall assessment:

UTIs is the main driver of antibiotic resistance and CAUTI is major problem in society and hospitals settings. Fundamental knowledge about UTI/CAUTI pathogenesis is lacking. The urinary catheter was invented roughly 100 years ago, yet today the conceptual principle and design is basically unchanged. Antimicrobial catheters have been explored for decades but no game-changing intervention have emerged. This is likely a result from missing insight in the CAUTI development pathways and the role of the urinary catheters. I strongly commend the authors efforts to address this important gap.

However, I do find the study appropriate for publication in its current state.

Most importantly, the manuscript is difficult to read with many unexplained terms. The results are poorly presented to a degree that spoils the quality of the work. This reviewer is not able to see how the conclusions of the manuscript can be drawn, because the results are basically not shown properly. It is not clear what are the results.

Furthermore, the supplementary material is too extensive with inappropriate amounts of bodies of text.

Finally, the manuscript seems to lack conjunction with the clinical reality.

This reviewer fails to comprehend what the study found, and the figures are very difficult to understand. Hence, it is difficult to see how the conclusions can be drawn. This reviewer represents a medical doctor in clinical microbiology with previous clinical experience in urology and research focus only in UTI and CAUTI. The mathematical formulas are outside my area of expertise, however, this reviewer should still be able to understand this work as it is well within my interest area. The reviewer represents a very relevant potential reader of this manuscript if it was published, but simply do not understand the results and conclusions. Clinical doctors, for whom this study would be relevant, will not be able to read and understand it. The work does not seem very substantial.

Specific comments:

According to the model, the authors identify a specific urine production rate of 1.16 mL/min to be predictive of whether a patient will develop bacteriuria or not. This makes sense, and could serve as a clinical guide for this patient group. The only concern is, whether this specific number can be extrapolated to real patients. The authors do not discuss how there findings may be used to the benefit of patients.

It seems that the model predicts bacteriuria detection time to be about 170 days. This does not really make sense, as indwelling catheters are only supposed to stay inserted for up to 90 days. And we know from clinical studies that bacteriuria is detected in almost all catheterized patients after 7-10 days. Similarly, in figure 5, only 50 and 60 days are shown which is a very long time given that bacteriuria develops much faster. These time frames are not in conjunction with the clinical reality. There are other examples of this.

The story is difficult to follow:

- Specific cross-references to supplementary material. This needs to be much more specific.
- There are a number of phases that are not explained sufficiently. E.g., steady-state outcome; bacterial volume density. The

authors describe high- and low bacterial density regimes. What is meant by this?

- There is a significant amount of discussion in the results section, making it difficult to identify what the actual results are.
- The study is not concise.
- This manuscript would benefit from line number to ease the reviewing process.

The supplementary material is far too extensive. The supplementary material must be reduced to a minimum and only include non-essential data otherwise it should be part of the main manuscript. E.g., the first three pages of the supplementary material is additional background information which is not appropriate to include as supplementary material.

The authors use extra- and intraluminal to describe the surface of the inside and outside bladder. Change to luminal and outside surface.

Page 2, first sentence. The authors should specify that long-term indwelling catheterization is up to 3 months – then renewal (and not life-long as it could be misread in its current form).

Page 7. Bacteria are not removed from the bladder by dilution (they are just being diluted) – only by urine flow will they actually be removed (and by active killing from the immune system).

A lot of the text should be removed from the results section (and discussed in the discussion). This would help to make the results more clear.

Page 8: “the model suggests that such patients could still experience outcomes such as catheter blockage, even without developing bacteriuria.”. I do not see how the authors reach this conclusion.

Page 9. What is steady-state outcome.

Page 10, line 3. Urethra length?

Page 11, line 2. Intracellular bacteria as a source of CAUTI is hypothetical. This is largely a mouse phenomenon and not evidenced in humans.

Page 11, line 4-5: This is not what is shown in the figure. The figure shows colonization only in certain areas of the catheter.

Page 11: “Finally, if the catheter becomes contaminated on insertion, such that bacteria become spread all over the extraluminal surface”. This would never happen as catheters are placed as sterile procedures. Contamination is definitely a concern, but to that extent.

Figure 2 legend. I don't understand the concentration of the inoculum. Does the author inoculated with 100 bacteria pr mm²?

Units are not consistent. mL/min; mL min⁻¹; mm³s⁻¹. This is confusing. I would recommend that authors decide on mL /min which is the typical reporting in medicine, but I understand the authors represents other fields with other traditions. Similarly, the authors use ‘bacterial density’ to describe the amount of bacteria in certain situations. It would increase the readability to use more conventional terms like bacteria/ml or bacteria/surface area.

Figure 4. What is the wash out regime? What is bacterial volume density (do they mean bacterial burden/colony counts?) I do not understand figure 4a: when are bacteria detected in the figure? For both panels, I think the figures would benefit from more ticks on both axis.

Figure 5. line 1, remove =. This figure and its legend is difficult to understand. What is the results /conclusions from this figure?

Table 2. This is a very long table legend with a reference to supplementary material – but what am I looking for in the supplementary material? What is missing?

(Remarks on code availability)

Reviewer #2

(Remarks to the Author)

This manuscript is a very well-written computational modeling study of bacterial colonization of a foley catheter that can predict many of the current observations related to development of bacteriuria in short- and long-term catheterization scenarios. Importantly, it gives a mechanistic description for the mixed results observed for various interventions. While models for micturition (non-catheterized) bacterial dynamics exist, this system adds the complexity of multiple colonization domains, namely intraluminal and extraluminal surfaces as well as the bladder and intraluminal fluid spaces. I see no fundamental issues with the implementation of their methods and assumptions. Their work suggests the need for a more nuanced approach to inventions to reducing CAUTI, rather than a one-size fits all which is impactful. Therefore, this work is worthy of publication. Some potential issues that could be addressed before final acceptance include:

The model does not account for the catheter balloon which has different material properties and therefore may have different effects on bacterial adhesion, colonization, growth, and biofilm formation. This should at least be mentioned as missing from the model.

Biofilm population dynamics is typically modeled as cyclical in that the bacterial adhere to the surface, proliferate until a threshold is reached, and then are released. It does not appear the modeling of the connections between the different colonization domains incorporates this quorum sensing phenomena. Could the authors provide at least a cursory description of how this might affect their results? As a corollary, bacteria can be shed from a surface through mechanical disruption via shear stress which results in flocs of biofilm material (larger than single cells) and with a different growth phenotype. How might this phenomena alter the role of urine production rate results?

Conservation of bacterial number at the interfaces of the domains could lead to inaccuracies. This assumption does not

account for bacterial proliferation that leads to dispersal and reattachment at another location.

There was a glaring lack of account of the effects of the immune system in killing and disposal of bacteria, which the authors mention. However, a more detailed list of the potential points in the model where it might impact is lacking.

Another neglected variable in the model is the nutrient content of the urine which would affect bacterial growth rates. The increased glucose in the urine of diabetics has been linked to their increased risk of urinary tract infection. This should be listed among other patient factors not accounted for including (immunosuppression, presence of renal disease, diuretic usage, etc).

The authors note that examinations of patterns of bacterial colonization on catheters at early times are not routinely performed. However, their work suggests that performing them may provide insight into where known interventions may be most beneficial and selectively applied. I think this concept could be amplified more.

Obstruction of the outflow of the Foley catheter may be related to debris from dead bacteria and or sloughing of epithelial cells from infection and/or hemorrhage related to infection and inflammation leading to clots. These additional clogging mechanisms should be mentioned as they may not equally contribute or necessarily be related to the number of bacteria.

(Remarks on code availability)

Reviewer #3

(Remarks to the Author)

In this paper the authors address the important question of the risk of bacterial colonization associated with urinary catheter. Specifically, they propose a novel mechanistic model to better understand bacterial colonization in different compartments of the catheter and the bladder. Using a large simulation study, they explore key parameters associated with high risk of bacterial invasion for long-term catheterization and short-term catheterization. Bacterial colonization and infection of catheters in individuals is a major public health problem that has been very poorly studied using mathematical models. By combining population dynamics and fluid theory, the authors propose here a new and very original modelling framework to shed light on this issue.

The article is clearly written (including the modelling part), the model is original and the simulations provide interesting results for a better understanding of the factors associated with colonization risk at the individual level. However, there is a lack of validation of the model and results with real data, which clearly limits the impact of the paper.

Comments:

- 1) Figures 1 and 2 provide a global illustration of the different layers of the model but they do not provide any information about the model. In fact, it does not provide any information on the assumptions and mechanisms included in each sub-model. Since the main originality of this article is actually the model, it would be much clearer for the reader to include, as a subpanel of this figure (or later in the "Material and Methods" section), a detailed schematic of the different mathematical mechanistic models developed for each component, defining the different variables and parameters that drive the dynamics. I would also be useful to include a schematic of the connection of the different parts (eg something similar to Figure S4 for each component connection)
- 2) Validation of models and confrontation with data. It is not clear to me to what extent the proposed modelling framework has been confronted with data at each layer for validation. My understanding is that there was no confrontation with data and no global sensitivity analysis. It would really strengthen the article and the results to provide some comparison of model outcomes with real colonization data. Would the in vitro experiments of catheterized bladder mentioned by the authors be useful data?
- 3) The authors present a large number of analyses of the effect of parameters for different outcomes such as the steady state, the time to detection of bacteriuria, the density of bacterial population in the catheter etc. Did you (and could you) do some global sensitivity analysis to address the question of the contribution of the different parameters of the different strata globally? For example, could you provide some quantitative impact of the different parameters in Table 2 instead of qualitative summaries?
- 4) "This is the first quantitative model of the dynamics of urinary catheter colonization, ..." : I agree with this statement and on the great interest of the model. However the lack of linkage of the model to real data hinders a real "quantitative" analysis and rather provides more qualitative explanations of global observations made in patients. Again, a proper confrontation with data for model parameterization would clearly strengthen the relevance and applicability of the results presented here.
- 5) "the model also suggests that individuals might reduce their susceptibility by increasing fluid intake..". This is an interesting application with public health implications. Could the authors provide more detailed and quantitative results on this point?
- 6) Has some stochasticity been included in the model. To what extent (and in what component) would some stochastic

processes affect the various risks and outcomes under consideration?

Minor comments:

7) The paragraph beginning “by simulating...” at the end of the introduction section, summarizes all the results obtained in the article. It seemed strange to me to find this at this stage of the article and would recommend moving it to the discussion section.

(Remarks on code availability)

Version 1:

Reviewer comments:

Reviewer #3

(Remarks to the Author)

First of all, I would like to congratulate the authors on the considerable effort that was made to answer all the reviewers' comments. I think the paper has been largely improved. A few remaining comments:

1) The authors made a very nice effort in including new analyses in which the model was compared to published data and parameters estimated, making the results, although still qualitative and theoretical, probably more linked to real experiments. The pity is that there is no possibility for the reader, in the current state, to really evaluate the adequation of the model to the cited data. It would be good if the authors could include a new figure illustrating such model agreement to data (or possible disagreement for range of parameters values other than the selected ones?).

2) Sensitivity analysis – could the authors provide some justification/explanation on the choice of the lower and upper bounds associated values in the table?

3) Figure formalizing the model. I recognize and acknowledge the efforts of the authors in providing an expanded schematic of the model on Figures 1 and 2. However, I am still not convinced with the current state of these figures: I still believe that it is hard for a reader specifically interested in the modeling aspects to really understand precisely the considered processes and assumptions. For example, parameters were added to figure 1 but we still lack a mechanistic representation of the modelled processes, eg growth, motility or urine impact. Although I agree that this paper targets an interdisciplinary range of readers, it presents a theoretical analysis - it is therefore very important to provide a detailed picture of the assumed processes at a fine scale. If the authors do not manage to better represent mechanisms, some terms of equations could be included as a last column of figures 1 and 2. This said, Figure S4 is a good example of what could be expected for such a figure in the main text (a figure properly integrating the processes).

(Remarks on code availability)

Response to the comments of the reviewers

Reviewer 1's comments

UTIs is the main driver of antibiotic resistance and CAUTI is major problem in society and hospitals settings. Fundamental knowledge about UTI/CAUTI pathogenesis is lacking. The urinary catheter was invented roughly 100 years ago, yet today the conceptual principle and design is basically unchanged. Antimicrobial catheters have been explored for decades but no game-changing intervention have emerged. This is likely a result from missing insight in the CAUTI development pathways and the role of the urinary catheters. I strongly commend the authors efforts to address this important gap.

We thank the reviewer for these very positive comments about our manuscript. We are delighted that the reviewer agrees with us on the importance of understanding CAUTI development pathways and commends our efforts to address the current gap in knowledge.

However, I do not find the study appropriate for publication in its current state. Most importantly, the manuscript is difficult to read with many unexplained terms. The results are poorly presented to a degree that spoils the quality of the work. This reviewer is not able to see how the conclusions of the manuscript can be drawn, because the results are basically not shown properly. It is not clear what are the results. Furthermore, the supplementary material is too extensive with inappropriate amounts of bodies of text. Finally, the manuscript seems to lack conjunction with the clinical reality. This reviewer fails to comprehend what the study found, and the figures are very difficult to understand. Hence, it is difficult to see how the conclusions can be drawn. This reviewer represents a medical doctor in clinical microbiology with previous clinical experience in urology and research focus only in UTI and CAUTI. The mathematical formulas are outside my area of expertise, however, this reviewer should still be able to understand this work as it is well within my interest area. The reviewer represents a very relevant potential reader of this manuscript if it was published, but simply do not understand the results and conclusions. Clinical doctors, for whom this study would be relevant, will not be able to read and understand it. The work does not seem very substantial.

We appreciate this helpful feedback on the need to make our manuscript more readable by a clinician. Our manuscript was written from a mathematical modelling perspective and, on reflection, we can see that some of the terms, and the presentation and clinical implications of the results, could have been clearer for a clinical reader. We are very keen to make our work accessible to clinicians since, as the reviewer states, our work should have significant clinical impact. Therefore we have made extensive changes to all sections of the text, with the aim of making the terms used more accessible, stating the key results more clearly and explaining more explicitly what we envisage as their clinical implications. We feel that these changes have substantially improved the readability of the manuscript and we hope the reviewer will agree. Changes are shown in red in the revised manuscript and also stated in detail in the response to specific comments below.

Specific comments

1. According to the model, the authors identify a specific urine production rate of 1.16 mL/min to be predictive of whether a patient will develop bacteriuria or not. This makes sense, and could serve as a clinical guide for this patient group. The only concern is, whether this specific number can be extrapolated to real patients. The authors do not discuss how there findings may be used to the benefit of patients.

We thank the reviewer for the suggestion to connect our predicted critical value for the urine production rate with a possible clinical recommendation. In response, we have calculated model predictions for the change in risk of bacteriuria (on long-term catheterisation) for males and females, associated with specific increases in fluid intake up to 500mL per day. These predictions are presented in a new table in the Supplementary Information (Table S1) and are also highlighted in the main text, first results section, where we now state “For example, we predict that increasing urine production rate by 300 mL/day (0.21 mL/min) would result in a decrease in relative risk of bacteriuria of 12.7 +/- 0.8 % (females) or 15.1 +/- 0.9 % (males) (see Supplementary Material IV.B, Table S1).” We feel that these predictions do indeed strengthen the usefulness of our work as a guide for clinicians.

It seems that the model predicts bacteriuria detection time to be about 170 days. This does not really make sense, as indwelling catheters are only supposed to stay inserted for up to 90 days. And we know from clinical studies that bacteriuria is detected in almost all catheterized patients after 7-10 days. Similarly, in figure 5, only 50 and 60 days are shown which is a very long time given that bacteriuria develops much faster. These time frames are not in conjunction with the clinical reality. There are other examples of this.

The reviewer is right, in the previous version of the manuscript our model predicted unrealistically slow catheter colonisation dynamics. The origin of this issue was the parameter value for the mobility of bacteria on the surface of the catheter, for which experimental measurements are not available. Previously we estimated this parameter by assuming that bacteria spread on the catheter by dividing and pushing each other apart. However, fitting of our model to measurements of the distribution of bacteria on catheters as a function of time in an in vitro model system (Zhang et al (Biomat. Sci. Eng 2019) and Wang et al (J. Hosp. Infect. 2019)) suggests a much higher bacterial motility on the catheter surface (see response to reviewer 3 below). This might hint that bacteria migrate up the catheter through a mucus layer surrounding the catheter. We also obtain similar results for the motility parameter in ongoing work where we fit our model to clinical data on the prevalence of bacteriuria in >7000 catheterised patients (Pickard et al, Lancet 2012).

We have now added a new section in the Supplementary Material (Section IV E) describing the fitting to the data of the Zhang et al and Wang et al study, and have changed the bacterial motility parameter to reflect our new understanding. This gives much more realistic values for the bacteriuria detection time, reflected in our updated version of Figure 4 “Time to detection of bacteriuria”.

2. The story is difficult to follow:

- Specific cross-references to supplementary material. This needs to be much more specific.

We have now added specific cross-references to the supplementary material. We agree that this makes the work easier to follow.

- There are a number of phases that are not explained sufficiently. E.g., steady-state outcome; bacterial volume density. The authors describe high- and low bacterial density regimes. What is meant by this?

The terms “steady-state outcome” and “bacterial volume density” are standard in the mathematical modelling field, but following the reviewer’s comment we now realise that they might not be clear to readers from a different background. Therefore we have changed the wording in most of the manuscript, only leaving these terms in the Materials and Methods where we feel they help to explain the technical aspects of the mathematics. For example, in the first results section we have replaced our statement about the high and low bacterial density regimes as follows: “We observe two possible outcomes. In one

outcome, which we associate with bacteriuria, the abundance of bacteria in the urine is high, while in the other outcome, which we associate with no bacteriuria, the abundance of bacteria in the urine is low." We thank the reviewer for pointing this out and we hope he/she will agree that the manuscript is now much easier to read.

- There is a significant amount of discussion in the results section, making it difficult to identify what the actual results are.

We appreciate the reviewer's point – our manuscript is indeed structured differently from a standard article in the clinical field. Inclusion of discussion points in the results section is more routine in mathematical modelling articles, and we prefer to keep it here, since we feel that it helps to explain the significance of our results, which could otherwise be difficult to grasp for a non-clinical reader. However, we have tried to address the reviewer's concern by adding extra sentences that state explicitly what the key results are.

- The study is not concise.

We appreciate this point but since we would like to make our work accessible to readers from very different backgrounds, from clinical to mathematical modelling and biophysics, we feel that we need to include more explanation/discussion that might normally be needed in a more specialised article.

- This manuscript would benefit from line number to ease the reviewing process.

Thanks for the suggestion, we have added line numbers.

3. The supplementary material is far too extensive. The supplementary material must be reduced to a minimum and only include non-essential data otherwise it should be part of the main manuscript. E.g., the first three pages of the supplementary material is additional background information which is not appropriate to include as supplementary material.

We appreciate that the supplementary material is long, but we feel it is important to recognize the very diverse readership of our work. While the background information in the first 3 pages is clearly not necessary for a clinical reader, it may well be new information for a biological physicist. We also feel that it is important to present the mathematical calculations in detail to allow a mathematical reader to fully assess the work. Therefore, we prefer to err on the side of presenting a too detailed supplement, rather than having more mathematical or physics-oriented readers be unable to follow it.

To make the supplement more accessible (despite its length) we have added a table of contents as well as more specific cross-references between the main text and supplement.

The authors use extra- and intraluminal to describe the surface of the inside and outside bladder. Change to luminal and outside surface.

This has been done.

Page 2, first sentence. The authors should specify that long-term indwelling catheterization is up to 3 months – then renewal (and not life-long as it could be misread in its current form).

Thanks, we have amended this in the introduction.

Page 7. Bacteria are not removed from the bladder by dilution (they are just being diluted) – only by urine flow will they actually be removed (and by active killing from the immune system).

We have clarified this point, amending the relevant sentence to “The urine production rate is a key parameter in our model, since it controls the rate at which bacteria are removed from the bladder in the urine flow; this corresponds to a dilution term in the mathematical model with rate k_D ”. From a mathematical point of view the production of urine corresponds to what is known as a “dilution term” in the equation, so we prefer not to remove the term “dilution” altogether, since this would make it hard for a modeller to understand the equations. We hope that what we have now written will be clear to diverse readers.

A lot of the text should be removed from the results section (and discussed in the discussion). This would help to make the results more clear.

We appreciate the reviewer’s point, but as discussed also above, we feel that non-clinical readers will benefit from an analysis of the results as they are presented, i.e. in the results section. Therefore we prefer to keep this format, but we have reworded the text to highlight more clearly what the results are.

Page 8: “the model suggests that such patients could still experience outcomes such as catheter blockage, even without developing bacteriuria.”. I do not see how the authors reach this conclusion.

We appreciate that this point was not clear and we have re-worded it to explain the mechanism more clearly. It now reads: “Hence, the presence of the catheter allows an infection to persist even if the urine production rate is high enough to wash it out of the bladder, since the infection can be “re-seeded” from the biofilm on the catheter surface (see Supplemental Material IV.C for further discussion). Indeed, catheter-associated biofilms are known to act as a bacterial reservoir, leading to persistent re-colonisation of the bladder [34]. Therefore our model predicts that even patients who do not experience bacteriuria (e.g. because they have a high urine production rate; Fig. 3b) may still experience adverse outcomes such as catheter blockage.”

Page 9. What is steady-state outcome.

Steady state outcome means the prediction of the model for long times, after its predictions have stopped changing in time. We now explain this explicitly, and we have also removed the term “steady-state” in most places, replacing it with “long-term”.

Page 10, line 3. Urethra length?

This has been fixed.

Page 11, line 2. Intracellular bacteria as a source of CAUTI is hypothetical. This is largely a mouse phenomenon and not evidenced in humans.

Thanks, we now state that this phenomenon is specific to mouse models.

Page 11, line 4-5: This is not what is shown in the figure. The figure shows colonization only in certain areas of the catheter.

This comment refers to the statement “different sources of infection lead to different patterns of bacterial density”. The reviewer is right that what we actually predict is that different parts of the catheter get colonised at different times depending on the infection pathway. We referred to this as “patterns” because it can also be that only part of, for example, the luminal surface, is predicted to be colonised at a given time, for example the lower part is colonised but not the upper part, and the colonisation proceeds as a wave that spreads in time. However, we appreciate that not everyone might recognise this as a “pattern”. We still feel that this is a useful way to think about the model prediction, so we have left some instances of the word “pattern” in this section of the manuscript, but we have removed this term where we think it is clearer to be more explicit. For example we now give an explicit clinical suggestion at the end of the section “For example, where a catheter has been removed due to infection, it could be inspected for bacterial colonisation of the lower and upper outside vs luminal surfaces, potentially informing about the most likely source of infection and informing targeted infection prevention measures for the next catheterisation.”

Page 11: “Finally, if the catheter becomes contaminated on insertion, such that bacteria become spread all over the extraluminal surface”. This would never happen as catheters are placed as sterile procedures. Contamination is definitely a concern, but to that extent.

We thank the reviewer for this insight. We feel that the case where bacteria are initially spread over the catheter is interesting for understanding how the model works, but we appreciate the reviewer’s point that it would imply a breach of sterile procedures. We now state this explicitly in the manuscript (last part of the results section).

Figure 2 legend. I don’t understand the concentration of the inoculum. Does the author inoculated with 100 bacteria pr mm²?

Yes this is what was meant, but we appreciate the notation was not clear. This has now been clarified.

Units are not consistent. mL/min; mL min⁻¹; mm³s⁻¹. This is confusing. I would recommend that authors decide on mL /min which is the typical reporting in medicine, but I understand the authors represents other fields with other traditions.

The reviewer is right, the use of multiple units is confusing. In places we have to use different units, since for example, the abundance of bacteria on a surface (bacteria per surface area) is a fundamentally different quantity from in a volume (bacteria per volume). But we were previously using two different units for urine flow rate (mm³ /s and mL /min). Following the reviewer’s suggestion we now use only mL/min throughout.

Similarly, the authors use ‘bacterial density’ to describe the amount of bacteria in certain situations. It would increase the readability to use more conventional terms like bacteria/ml or bacteria/surface area.

We agree and we have removed the term “bacterial density” in most places in the manuscript, only keeping it in the Materials and Methods where we present the equations, since it is a standard terminology in mathematical modelling.

Figure 4. What is the wash out regime? What is bacterial volume density (do they mean bacterial burden/colony counts?) I do not understand figure 4a: when are bacteria detected in the figure? For both panels, I think the figures would benefit from more ticks on both axis.

The washout regime is a term used in the chemostat literature, where bacteria are grown in a vessel that is continuously diluted with fresh media. “Washout” happens when the dilution rate is so high that bacteria cannot grow fast enough to maintain a stable population in the chemostat. Here we make the point that our bladder model is similar to a chemostat and that the “no bacteriuria” regime of our model is similar to the washout regime of a chemostat. We now try to make this clearer in the text: “The bacteriuria transition predicted by our model is similar to the washout phenomenon that occurs in continuous bacterial culture, i.e. in a chemostat, when the dilution rate approaches the bacterial maximal growth rate, so that the bacteria in the chemostat device cannot grow fast enough to keep up with dilution and are “washed out” of the device.”

Figure 4a plots the time after infection at which our model predicts bacteriuria will be detected. After infection, the abundance of bacteria in the urine increases and eventually reaches an assumed detection threshold: here we plot the time at which the detection threshold is crossed. We have now clarified this in the text with the following sentence: “To assess the short-term outcome, we track in our computer simulations the time after infection at which bacterial abundance in the urine reaches a threshold, which we denote the “detection threshold for bacteriuria” (Fig. 4)”

Figure 4a and 4b have changed since we now use an improved parameter value for the bacterial mobility on the surface. They both have more ticks on the axes than in the previous version.

Figure 5. line 1, remove =. This figure and its legend is difficult to understand. What is the results /conclusions from this figure?

The typo has been fixed. We have rewritten the legend so that hopefully it is easier to follow. The key result is that the pattern of bacterial abundances on the outside and luminal catheter surfaces differs for different infection scenarios. This point is now made explicit in the title of the figure.

Table 2. This is a very long table legend with a reference to supplementary material – but what am I looking for in the supplementary material? What is missing?

We have shortened the legend of table 2, added specific references to the relevant parts of the supplementary material and explained what is referred to in the supplement: “... for a detailed discussion of how the model behaves differently depending on the bacterial motility on the catheter surface”.

Reviewer 2’s comments:

This manuscript is a very well-written computational modeling study of bacterial colonization of a foley catheter that can predict many of the current observations related to development of bacteriuria in short- and long-term catheterization scenarios. Importantly, it gives a mechanistic description for the mixed results observed for various interventions. While models for micturition (non-catheterized) bacterial dynamics exist, this system adds the complexity of multiple colonization domains, namely intraluminal and extraluminal surfaces as well as the bladder and intraluminal fluid spaces. I see no fundamental issues with the implementation of their methods and assumptions. Their work suggests the

need for a more nuanced approach to interventions to reducing CAUTI, rather than a one-size fits all which is impactful. Therefore, this work is worthy of publication.

We thank the reviewer for these very encouraging comments. We are glad that the reviewer finds our manuscript well-written and appreciates the importance of providing a mechanistic picture of the process of bacterial colonization of a catheter, as well as the clinically relevant outcomes of the model. We find the reviewer's comment that our work suggests a more nuanced approach to interventions especially helpful, therefore we have modified the last line of the discussion to make this point explicitly.

Some potential issues that could be addressed before final acceptance include:

1. The model does not account for the catheter balloon which has different material properties and therefore may have different effects on bacterial adhesion, colonization, growth, and biofilm formation. This should at least be mentioned as missing from the model.

We thank the reviewer for this and the following suggestions. We have now added a new paragraph in the Discussion that covers this point as well as the other suggestions. This reads (with references removed for clarity):

"Our model could be extended in future to bring it closer to clinical reality. To account for the consequences of immune activation on bacterial dynamics and host response one should combine our model with existing immune system models -- this would allow, for example, predictions of fever symptoms and differences in CAUTI progression for immunosuppressed patients. It would also be interesting to model biofilm growth on the catheter surfaces in more detail, including for example the role of quorum sensing in biofilm initiation, triggered biofilm dispersal and the interplay between biofilm structure and fluid flow in the catheter lumen. It may be especially interesting to investigate how bacteria that disperse from biofilm (either as single cells or aggregates) on one part of the catheter surface could seed new biofilms on other parts of the catheter, potentially accelerating catheter blockage. It would also be interesting to investigate how a developing biofilm in the lumen alters the urine flow pattern, in turn altering the patterns of bacterial deposition on the lumen surface and potentially accelerating catheter blockage. However to properly model catheter blockage one should also include not only the possibility of crystalline biofilm formation due to alkalisation of the urine by *Proteus mirabilis*, but also the possible role of debris from dead bacteria, sloughed epithelial cells and/or blood clots. We have also ignored the possibility of biofilm growth on the catheter balloon, whose different material properties might affect biofilm formation and growth dynamics. Our modelling of bacterial dynamics in the bladder could also be extended to account for patient-to-patient differences in nutrient availability, most obviously the increased glucose concentration in diabetic patients, which has been linked to increased risk of urinary tract infection. Urine is a complex medium for which detailed bacterial growth models are lacking -- development of such models would be a useful direction for future research."

2. Biofilm population dynamics is typically modeled as cyclical in that the bacterial adhere to the surface, proliferate until a threshold is reached, and then are released. It does not appear the modeling of the connections between the different colonization domains incorporates this quorum sensing phenomena. Could the authors provide at least a cursory description of how this might affect their results? As a corollary, bacteria can be shed from a surface through mechanical disruption via shear stress which results in flocs of biofilm material (larger than single cells) and with a different growth phenotype. How might this phenomena alter the role of urine production rate results?

The new paragraph that we have added in the discussion (see answer to point 1 above) covers this point and also raises the interesting question of how a biofilm that develops in the lumen might alter the flow profile, accelerating further bacterial deposition and hence speeding up catheter blockage. We note however that detailed modeling of the biofilm formation process, while it would be extremely interesting, would mainly relate to how the catheter blocks (ie the later stages of catheter colonisation), not to whether or how fast bacteriuria develops (ie the earlier stages of catheter colonisation). Therefore we do not expect that the main qualitative results presented in our paper would be affected by more detailed modelling of the biofilm.

3. Conservation of bacterial number at the interfaces of the domains could lead to inaccuracies. This assumption does not account for bacterial proliferation that leads to dispersal and reattachment at another location.

This is a misunderstanding - “conservation of bacterial number” is a technical mathematical term that on reflection we realise could be confusing to a biological or clinical reader. In fact the way our model deals with the connection between model parts does correctly account for proliferation of bacteria. To focus on the example suggested by the reviewer, bacteria could proliferate as a biofilm, for example on the outer catheter surface, that then disperses, and the bacteria could then be transferred to the bladder or to the luminal surface of the catheter. The equations that deal with connection between the model parts merely keep track of bacteria that move from one part of the model to another – it does not place any restrictions on proliferation, dispersal or reattachment. To avoid confusion on this point we have removed the statement “in each case, the number of bacteria is conserved” (for mathematical readers, we feel this point will anyway be clear from the equations).

There was a glaring lack of account of the effects of the immune system in killing and disposal of bacteria, which the authors mention. However, a more detailed list of the potential points in the model where it might impact is lacking.

The reviewer is of course right that the immune system is important and is neglected in our model. This point is now discussed in more detail in our new paragraph in the discussion (see answer to point 1 above), where we note that it might impact not only bacterial growth and killing dynamics but would also allow prediction of host clinical symptoms and different outcomes in immunocompromised patients. We also explain that properly including the immune system in the model would require coupling to a detailed model of the immune system, giving references to some such models. This is, however, certainly beyond the scope of the present work.

Another neglected variable in the model is the nutrient content of the urine which would affect bacterial growth rates. The increased glucose in the urine of diabetics has been linked to their increased risk of urinary tract infection. This should be listed among other patient factors not accounted for including (immunosuppression, presence of renal disease, diuretic usage, etc).

We thank the reviewer for bringing up this important point. Actually when developing our model we originally planned to include a more detailed description of the bacterial growth dynamics in the urine, accounting for nutrient content, but we found no suitable theoretical models for growth in urine that we could use. Therefore we decided to use the simplified logistic growth form that we present in the manuscript. We do agree, however, that future work should focus on the question of how bacteria grow

in urine, with the aim of including this in more detail in extended versions of our model. This point is now made in the added paragraph in the discussion section (see answer to point 1 above).

The authors note that examinations of patterns of bacterial colonization on catheters at early times are not routinely performed. However, their work suggests that performing them may provide insight into where known interventions may be most beneficial and selectively applied. I think this concept could be amplified more.

We are glad that the reviewer finds our predictions about the patterns of bacterial colonisation on catheters potentially clinical useful, and we thank him/her for the suggestion to amplify this point. In response we have added at the end of the Results section the following text: "In addition, this approach could be helpful for optimising catheter management protocols. For example, where a catheter has been removed due to infection, it could be inspected for bacterial colonisation of the lower and upper outside vs luminal surfaces, potentially informing about the most likely source of infection and informing targeted infection prevention measures for the next catheterisation."

Obstruction of the outflow of the Foley catheter may be related to debris from dead bacteria and or sloughing of epithelial cells from infection and/or hemorrhage related to infection and inflammation leading to clots. These additional clogging mechanisms should be mentioned as they may not equally contribute or necessarily be related to the number of bacteria.

We thank the reviewer for pointing this out. While we do not focus in detail on catheter blockage in our manuscript, modelling blockage in detail would be an obvious extension of the model and we now comment on the possible contributions of dead bacteria, epithelial cells and blood clots to blockage in the new paragraph that we have added in the discussion section (see response to point 1 above).

Reviewer 3's comments:

In this paper the authors address the important question of the risk of bacterial colonization associated with urinary catheter. Specifically, they propose a novel mechanistic model to better understand bacterial colonization in different compartments of the catheter and the bladder. Using a large simulation study, they explore key parameters associated with high risk of bacterial invasion for long-term catheterization and short-term catheterization. Bacterial colonization and infection of catheters in individuals is a major public health problem that has been very poorly studied using mathematical models. By combining population dynamics and fluid theory, the authors propose here a new and very original modelling framework to shed light on this issue. The article is clearly written (including the modelling part), the model is original and the simulations provide interesting results for a better understanding of the factors associated with colonization risk at the individual level. However, there is a lack of validation of the model and results with real data, which clearly limits the impact of the paper.

We are glad that the reviewer finds our manuscript clearly written and the results interesting, and that the study addresses an important knowledge gap. We understand the reviewer's concern about the lack of validation with real data. We have addressed this concern in part by including a comparison with data from Zhang et al (Biomat. Sci. Eng 2019) and Wang et al (J. Hosp. Infect. 2019), who measured the density profiles of bacteria on catheters for an in vitro model system in which a catheter was embedded in an agar 'urethra' with an artificial urine growth medium flowing down the catheter from a model bladder. This comparison now forms a new section (IV E) in the Supplementary Material. We are also in the process of carrying out a more extensive test of our model using the clinical dataset from the study

of Pickard et al (Lancet 2012), who assessed the incidence of bacteriuria in over 7000 patients who were catheterized post-surgery. However, as we explain below, we feel that this work requires its own paper and would overload the current manuscript.

1. Figures 1 and 2 provide a global illustration of the different layers of the model but they do not provide any information about the model. In fact, it does not provide any information on the assumptions and mechanisms included in each sub-model. Since the main originality of this article is actually the model, it would be much clearer for the reader to include, as a subpanel of this figure (or later in the “Material and Methods” section), a detailed schematic of the different mathematical mechanistic models developed for each component, defining the different variables and parameters that drive the dynamics. I would also be useful to include a schematic of the connection of the different parts (eg something similar to Figure S4 for each component connection)

We thank the reviewer for this suggestion and have updated figures 1 and 2. In figure 1 we now describe all of the physical processes within the model, as well as annotating the “top”/“base” of the catheter. In figure 2 we have added a list of the key physical parameters and corresponding notation for each section of the model.

2. Validation of models and confrontation with data. It is not clear to me to what extent the proposed modelling framework has been confronted with data at each layer for validation. My understanding is that there was no confrontation with data and no global sensitivity analysis. It would really strengthen the article and the results to provide some comparison of model outcomes with real colonization data. Would the in vitro experiments of catheterized bladder mentioned by the authors be useful data?

The reviewer’s point about global sensitivity analysis is addressed below (point 3). Regarding confrontation with data, we certainly agree that this is important. Following the reviewer’s suggestion, we now compare our model directly with a dataset of bacterial density profiles on silicone and silver-PTFE coated catheters in an in vitro bladder model, in which a catheter was embedded in an agar ‘urethra’ with an artificial urine growth medium flowing down the catheter from a ‘bladder’. Zhang et al (2019) report the biofilm density (via UV absorbance after crystal violet staining) on catheter sections at different positions along the catheter similar to the predictions in Figure 5 of our manuscript. They also report the time from inoculation until the bacterial density in the “bladder” reached 10^3 CFU/mL. In a second paper from the same study, Wang et al (2019) compare results for several inoculation densities of bacteria at the base of the catheter.

Encouragingly, we note that the shapes of the patterns of bacterial density reported in these studies are strikingly similar to the wavelike shapes that are predicted by our model. More quantitatively, from this data we can estimate the parameters r_s and D_s in our model for silver-PTFE catheters, compared to silicone catheters. We see that the parameter r_s , that accounts for net bacterial growth on the catheter surface, is much smaller for the silver-PTFE surface than the silicone surface, which makes sense since silver should have antimicrobial efficacy. However, the bacterial motility, reflected in the parameter D_s , is larger for the silver-PTFE surface. Our parameter fitting also suggests that the bacterial strain used in this study (an uncharacterised clinical isolate) was unusually motile and unusually slow-growing compared to the usual growth rate of bacteria in urine. However the artificial urine medium that was used in the study might have constrained the growth rate.

We now report this comparison with data in detail in an additional section (IV E) in the Supplementary Material, and reference it also from the main text.

As noted above, we are also in the process of comparing our model to data from the clinical study of Pickard et al (Lancet 2012), who assessed the incidence of bacteriuria in over 7000 patients who were catheterized post-surgery with PTFE, nitrofurantoin-impregnated or silver-coated latex catheters. Preliminary results suggest that our model fits the data quite well for the dependence of bacteriuria incidence on duration of catheterisation. However, this analysis is not yet complete and is complicated by various factors including that some patients had pre-existing bacteriuria, some were antibiotic treated and some not, making it a research study of its own. Hence we prefer not to include it in the present manuscript but rather to devote a separate, future paper to it, where it can be presented properly, with the appropriate level of detail and statistical rigour.

3. The authors present a large number of analyses of the effect of parameters for different outcomes such as the steady state, the time to detection of bacteriuria, the density of bacterial population in the catheter etc. Did you (and could you) do some global sensitivity analysis to address the question of the contribution of the different parameters of the different strata globally? For example, could you provide some quantitative impact of the different parameters in Table 2 instead of qualitative summaries?

We thank the reviewer for this helpful suggestion. In response we have performed a global parameter sensitivity analysis to quantify the sensitivity of two outcomes: the steady state bacterial abundance in the bladder and the time taken to reach 99% of this steady state abundance (starting from the uninfected state), to the 10 parameters listed in Table 2 of the main text. 360,000 parameter combinations were sampled using a Sobol sequence. We determined using Sobol indices that the key parameters controlling the steady state abundance in the bladder are, as expected, different than those that control the time to reach steady state. These findings are in good agreement with the original insights that are presented in our main results sections.

We have added a new section in the supplementary material (IV F) including a table of Sobol indices and 2 new supplementary figures. We also refer to this new analysis in the main text. The code for the parameter sensitivity analysis has been made publicly available via our github link.

4. "This is the first quantitative model of the dynamics of urinary catheter colonization, ..." : I agree with this statement and on the great interest of the model. However the lack of linkage of the model to real data hinders a real "quantitative" analysis and rather provides more qualitative explanations of global observations made in patients. Again, a proper confrontation with data for model parameterization would clearly strengthen the relevance and applicability of the results presented here.

We are very glad that the reviewer agrees with our assessment of the novelty and interest of the model. As explained above, we share the reviewer's view that it is important to link our model to real data, and we have fitted our model to the data of Zhang et al (2019) and Wang et al (2019) in the revised version, with an additional section in the Supplementary Material (IV E). As also explained above, we are also working on a more extensive comparison of our model to the clinical data of Pickard et al (2012), but we feel that this needs to be presented in its own, future, manuscript.

5. "the model also suggests that individuals might reduce their susceptibility by increasing fluid intake..". This is an interesting application with public health implications. Could the authors provide more detailed and quantitative results on this point?

We thank the reviewer for this interesting suggestion that has motivated us to use our model to quantify the extent of bacteriuria risk reduction associated with a given increase in urine production rate. It is important to note that we cannot predict directly the effects of increasing fluid intake, since our model only includes urine production rate as a parameter, it does not model how fluid intake leads to urine production, which may differ between different patients. Notwithstanding this caveat, our model suggests that increasing urine production rate by 300 mL/day would result in a decrease in relative risk of bacteriuria of 12.7 +/- 0.8 % (females) or 15.1 +/- 0.9 % (males). This is now stated in the main text (first results section) and explained in more detail in the Supplementary Material section IV.B, Table S1.

6. Has some stochasticity been included in the model. To what extent (and in what component) would some stochastic processes affect the various risks and outcomes under consideration?

The reviewer is right to point out that our model is deterministic – we have not included stochasticity in either the bacterial growth and motility dynamics or the initial arrival of bacteria the catheter. These factors could of course be included in future versions of the model, but actually we believe that the major sources of variability are more likely to be differences between the characteristics of different patients (affecting urethral length and urine production rate) and among different bacterial strains (affecting growth and motility parameters). This variability is, to some extent, taken into account in our manuscript since we compare outcomes for different urethral lengths over the typical range between male and female, and based on measured variability in urine production rate we comment on the fraction of the population that we expect to be susceptible to bacteriuria. We also investigate varying the bacterial mobility to mimic different bacterial strains. In the face of all this variability in patients and infecting strains, i.e. in the parameter values of the model, it would be interesting to understand how much additional contribution might arise from considering inherent stochasticity. In response to the reviewer’s comment we have added an additional paragraph on this topic in the discussion section.

7. The paragraph beginning “by simulating...” at the end of the introduction section, summarizes all the results obtained in the article. It seemed strange to me to find this at this stage of the article and would recommend moving it to the discussion section.

We appreciate the reviewer’s point here. Structuring our manuscript is challenging because we aim to make it accessible to biophysicists, microbiologists and clinicians, who naturally have different expectations about how a paper should be structured (this was also apparent in some of reviewer 1’s comments). We feel that summarizing the main conclusions at the end of the introduction has the advantage of making the paper accessible to readers who might not want to wade through the details of the modelling, as well as the disadvantage that it may seem strange to some readers. On balance we feel that accessibility is more important, so we prefer to leave the structure as it is. However, we have made extensive changes to the wording of the introduction, including in this paragraph, to make it clearer for clinical readers.

Response to reviewers

Reviewer #3 (Remarks to the Author):

First of all, I would like to congratulate the authors on the considerable effort that was made to answer all the reviewers' comments. I think the paper has been largely improved.

We thank the reviewer for their very positive comments. We agree that the paper has improved.

A few remaining comments:

1) The authors made a very nice effort in including new analyses in which the model was compared to published data and parameters estimated, making the results, although still qualitative and theoretical, probably more linked to real experiments. The pity is that there is no possibility for the reader, in the current state, to really evaluate the adequation of the model to the cited data. It would be good if the authors could include a new figure illustrating such model agreement to data (or possible disagreement for range of parameters values other than the selected ones?).

We thank the reviewer for their appreciation of our new analyses. We share the reviewer's awareness that it is important to evaluate the adequation of the model to the data. However, in this case, the experimental data is limited to measurements of the speed of the bacterial wave moving up the catheter for two cases, which we use to determine the parameter values of our model. Since only two numbers are involved, it is not possible to evaluate the goodness of the fit, or to plot a graph. As mentioned in our previous response, we are currently working on an analysis of a much more detailed dataset, that of Pickard et al (ref 24 in the revised manuscript). This analysis, which will be presented in a future paper, will allow us to plot the incidence of bacteriuria as a function of duration of catheterisation, comparing our model to data.

2) Sensitivity analysis – could the authors provide some justification/explanation on the choice of the lower and upper bounds associated values in the table?

We thank the reviewer for this suggestion and we have added a justification of the choice of bounds to Table S2.

3) Figure formalizing the model. I recognize and acknowledge the efforts of the authors in providing an expended schematic of the model on Figures 1 and 2. However, I am still not convinced with the current state of these figures: I still believe that it is hard for a reader specifically interested in the modeling aspects to really understand precisely the considered processes and assumptions. For example, parameters were added to figure 1 but we still lack a mechanistic representation of the modelled processes, eg growth, motility or urine impact. Although I agree that this paper targets an interdisciplinary range of readers, it presents a theoretical analysis - it is therefore very important to provide a detailed picture of the assumed processes at a fine scale. If the authors do not manage to better represent mechanisms, some terms of equations could be included as a last column of figures 1 and 2. This said, Figure S4 is a good example of what could be expected for such a figure in the main text (a figure properly integrating the processes).

We appreciate the reviewer's point and we share his/her concern that Figures 1 and 2 should be as clear as possible. However we do not see how we could illustrate the mechanisms involved graphically in more detail than we already have. That is because our model parameters are

general coarse-grained effective parameters that result from the combination of multiple underlying processes, which may vary depending on bacterial species. For example the motility parameter describes spreading of bacteria on the surface, that could happen by several different mechanisms. This approach is powerful because it is general: the model may be extended across bacterial species by parameterising with experimental measurements of the effective parameters, without needing detailed knowledge of specific processes. However the mechanisms are generic, and population-level, and do not allow more detailed illustration than the sketches that are already in Figures 1 and 2. The referee mentions figure S4, which does indeed give a more detailed sketch, but this is quite different because figure S4 demonstrates the technical geometry of how we consider the coupling between the top of the catheter and the bladder – it is a geometric illustration not one of a biological mechanism.

Following the reviewer's suggestion we also considered including more of the equations into Figure 2. However there is not space here to present the equations in their entirety and we feel that it would be misleading to present aspects of the equations in isolation, since the full set of equations is needed to understand the model. By presenting the model in the Methods, as we do now, we are able to present the full equations in one place: Equations 1 – 11.

Although we do not feel that we can add to the illustrations or include the model equations, we do appreciate the reviewer's point, and we have responded to it by adding text to figure 2 that provides brief descriptions of the modelling approach for each section of the model. We feel that this does make the figure clearer.